# Cecytb-2, a Cytochrome *b*_561_ Homolog, Functions as an Ascorbate-Specific Transmembrane Ferric Reductase at Intestinal Lumens of *Caenorhabditis elegans*

**DOI:** 10.3390/biom15101385

**Published:** 2025-09-29

**Authors:** Masahiro Miura, Misaki Fukuzawa, Hiroshi Hori, Kazuo Kobayashi, Mariam C. Recuenco, Motonari Tsubaki

**Affiliations:** 1Department of Chemistry, Graduate School of Science, Kobe University, Nada-ku, Kobe 657-8501, Hyogo, Japan; miura.masahiro@sysmex.co.jp (M.M.); misakifkzw@gmail.com (M.F.); hori2epr@trad.ocn.ne.jp (H.H.); 2The Institute of Scientific and Industrial Research, Osaka University, Mihogaoka 8-1, Ibaraki 567-0047, Osaka, Japan; kobayasi@sanken.osaka-u.ac.jp; 3Institute of Chemistry, College of Arts and Sciences, University of the Philippines Los Baños, Laguna 4031, Philippines; mcrecuenco@up.edu.ph; 4Research Facility Center for Science and Technology, Kobe University, Nada-ku, Kobe 657-8501, Hyogo, Japan

**Keywords:** ascorbate, ascorbate-specific electron transfer, cytochrome *b*_561_, *C. elegans*, Dcytb, ferric reductase, redox potential

## Abstract

One of the cytochrome *b*_561_ family members in *C. elegans*, named Cecytb-2, was investigated. Purified recombinant Cecytb-2 showed typical visible absorption spectra, EPR signals, and redox midpoint potentials, very similar to those of human Dcytb, which is responsible for intestinal iron acquisition by its ferric reductase activity. Fast kinetic experiments using pulse radiolysis and stopped-flow techniques showed that Cecytb-2 donates electrons to monodehydroascorbate radicals with a much lower reactivity than other cytochrome *b*_561_ members, but it can accept electrons from ascorbate (AsA) as rapidly as other members. DEPC treatment of Cecytb-2 caused significant inhibition of electron acceptance from AsA and lowered the midpoint potential of heme *b*_L_. MS/MS MASCOT analyses verified that *N*-carbethoxylations of conserved Lys98 and heme *b*_L_ axial His101 residues on the cytosolic side were major causes of the inhibition. Reconstituted Cecytb-2 in sealed vesicle membranes, in which AsA was entrapped, showed significant transmembrane ferric reductase activity. In situ hybridization analysis revealed that Cecytb-2 mRNA was distributed in intestinal cells. Immunohistochemical analysis indicated that Cecytb-2 resided in intestinal lumens. Knockdown of the Cecytb-2 gene expression in N2 worms indicated a significant suppression of growth under ferrous ion-deficient conditions. Thus, the ferric reductase activity conferred by Cecytb-2 seems to participate in iron acquisition and is very important for normal growth in low-ferrous conditions, confirming that Cecytb-2 is a genuine Dcytb homolog in *C. elegans*.

## 1. Introduction

Cytochrome b_561_ was found initially in adrenal chromaffin granules as a transmembrane electron transporter that contains two b-type heme prosthetic groups [1]. This protein (CGcytb) functions as a regenerator of ascorbic acid (AsA) inside the neuroendocrine vesicles of neural cells [2]. The AsA entrapped in neuroendocrine vesicles is consumed by copper-containing monooxygenases, such as dopamine β-hydroxylase and peptidylglycine α-amidating monooxygenase, resulting in the production of its one-electron oxidized form, monodehydroascorbate (MDA) radicals. It is thought that the regeneration of AsA from MDA radicals by CGcytb is an essential process for the normal production of neurotransmitters and their maintenance within vesicles [3,4], since neither AsA nor MDA radicals can pass through biomembranes without help from their specific transporters, which are absent in neuroendocrine vesicles.

Recent progress in various genome projects has revealed that a group of membrane proteins from various organisms, including vertebrates, insects, plants, and fungi, has a similar homology to CGcytb proteins [5]. These membrane proteins are classified as the cytochrome *b*_561_ protein family. It has been revealed that humans have six family members (i.e., adrenal (neural) cytochrome *b*_561_, CGcytb; duodenal cytochrome *b*, Dcytb; lysosomal cytochrome *b*, Lcytb; tumor suppressor cytochrome *b*, 101F6; stromal cell-derived receptor 2, SDR-2; and CYB561D1) [5,6]. Recent studies have shown that dysfunction or dysregulation of these family members is associated with various diseases. Pathogenetic mutations in CGcytb can lead to orthostatic hypotension [7], and a deficiency of CGcytb can cause selective sympathetic noradrenergic failure, leading to lifelong disabling orthostatic hypotension [8]. Furthermore, CGcytb promotes breast cancer growth [9,10] and knockdown of CGcytb inhibits the proliferation and migration of liver hepatocellular carcinoma cells [11]. Dcytb facilitates the progression of esophageal adenocarcinoma [12], but overexpression of Dcytb can impair lung adenocarcinoma cell proliferation, invasion, and adhesion [13]. Lcytb is crucial in the proliferation of Burkitt’s lymphoma cells [14], and SDR-2 accelerates cervical squamous cell carcinoma growth [15]. In addition, 101F6 represses the growth of human lung cancer cells [16] but contributes to the growth of glioma cells [15]. At present, the roles of these family members in tumors and their molecular mechanisms are still not clear, but the involvement of their ferric reductase activities [17] conferred by AsA-dependent transmembrane electron transfer [18] through ferroptosis might be considered. Ferroptosis is a newly defined form of regulated cell death characterized by excessive iron accumulation, iron-dependent lipid peroxidation, and subsequent cell membrane damage [19,20]; ferroptosis is likely to play a dual role in tumorigenesis, including promoting and inhibiting tumors [20].

In mammals, dietary iron is transported by divalent metal transporter-1 (DMT-1) residing in the apical surface of duodenal epithelial cells [21]. Because DMT-1 can transport only divalent ions (i.e., ferrous ions), the ferric ions in the lumen of the duodenal intestine must be reduced to a ferrous state in advance. Accordingly, it has been proposed that Dcytb acts as a transmembrane ferric reductase and catalyzes the reduction of extracellular ferric iron using electrons from cytosolic AsA [22,23].

The recent success of an X-ray crystallographic study on human Dcytb (hDcytb) revealed that each monomer of the homodimer protein possesses cytoplasmic and apical (or luminal) heme groups, as well as cytoplasmic and apical AsA-binding sites located adjacent to each heme [24]. It was further found that Zn^2+^ coordinates to two hydroxyl groups of the bound apical AsA and to a histidine residue (His108) and, accordingly, this site was proposed as a possible ferric ion-binding site for reductase activity [24].

Although a study using a *Dcytb* gene knockout mouse model raised the possibility that Dcytb is not essential for iron absorption in mice [25], later studies concluded that Dcytb is the primary iron-regulated duodenal ferric reductase in the gut and that Dcytb is necessary for optimal iron metabolism [26,27]. To elucidate the physiological functions of the diverse cytochrome *b*_561_ family members in animals, in the present study, we focused on a model organism, the nematode *Caenorhabditis elegans*, considering the presence of seven members of the family protein and its many advantages by using various analytical methods (physiological, biological, and molecular biological), as described in the previous literature (see recent reviews, such as [28]; i.e., inexpensive/easy to grow; genome is available; straightforward genetic tools exist; short generation time; short life span; small and exact 959 somatic cells; invariant development; transparent; organs and differentiated tissues; mutants can be frozen; homologous genes to human genes). Recent studies proved that *C. elegans* can biosynthesize ascorbate (AsA) [29] with a biosynthetic pathway similar to that found in mammals [30]. In our previous studies on cytochromes *b*_561_ in *C. elegans*, we reported that one homolog (named Cecytb-2) close to human Dcytb acted as a genuine ferric reductase, and its activity became enhanced upon reconstitution into a phospholipid bilayer nanodisc [31]. Here, we report detailed analyses on the biochemical and biophysical properties of the Cecytb-2 protein and its physiological functions as an AsA-specific transmembrane ferric reductase.

## 2. Materials and Methods

### 2.1. C. elegans Strains and Maintenance

The wild-type *C. elegans* strain Bristol N2, provided by the Caenorhabditis Genetics Center, was maintained on nematode growth medium (NGM) agar plates and fed with *E. coli* OP50 cells at 20 °C, as described by Brenner [32].

### 2.2. DNA Preparations

Total RNA was isolated from an adult *C. elegans* using the illustra RNAspin Mini RNA isolation Kit (GE Healthcare Japan, Hino, Japan). A *C. elegans* cDNA library was constructed by reverse transcription using a PrimeScript^®®^ II 1st strand cDNA Synthesis Kit (Takara Bio Inc., Kusatsu, Japan) with oligo dT primer. The full-length Cecytb-2 gene was amplified by PCR using the *C. elegans* cDNA library as a template. Two oligonucleotide primers, a forward primer (CCCTGCAGATGGAAGAGGAACAACTTCTTG, containing the *Eco*R I restriction site) and a reverse primer (ATGGATCCTCAATTGGTTCGGCCTTCG, containing the *Xho*I restriction site) were used to amplify the Cecytb-2 gene. The PCR product was digested with *Eco*RI and *Xho*I and then purified by a QIAquick PCR Purification Kit (QIAGEN K.K., Tokyo, Japan). This digested DNA fragment was inserted into the *Eco*RI/*Xho*I site of a pBlueScript II KS (+) vector. The resultant plasmid pBlueScript II KS (+)/Cecytb-2 was transformed into *E. coli* XL1Blue competent cells. The inserted DNA sequence was confirmed by DNA sequencing.

For the heterologous expression of the Cecytb-2 protein, we employed a *Pichia pastoris* expression system (EasySelect Pichia Expression Kit, Life Technologies Japan, Tokyo, Japan), as previously described [18,33,34,35]. The Cecytb-2 gene (*F39G3.5* gene) (Appendix A) was amplified by PCR using two oligonucleotide primers (a forward primer AAAGAATTCGCCACCATGTCTTCAGACTCTCGACTCGGCAATGCTC, containing an *Eco*RI site, and a reverse primer GGTCTCGAGTCATTAATGATGATGATGATGATGTGCGGACTTTAGCTCATCTGGAGTT, containing an *Xho*I site), with the purified pBlueScript II KS (+)/Cecytb-2 plasmid as a template. The reverse primer was designed to add a 6xHis-tag sequence at the C-terminus of the expressed Cecytb-2 protein. The recombinant protein, therefore, had a linker sequence (Ser-Ala) followed by a 6xHis-tag sequence fused at the C-terminus of the native Cecytb-2 protein (Appendix A). The PCR product was introduced into the *Eco*RI/*Xho*I site of the pPICZB vector by ligation. The resultant plasmid was transformed into *E. coli* XL1Blue competent cells. Then, the plasmid pPICZB-Cecytb-2-H_6_ was isolated, and the Cecytb-2 gene DNA sequence was confirmed by DNA sequencing. Finally, it was linearized by treatment with *Pme*I. The linearized plasmid was transformed into *P. pastoris* GS115 competent cells and was integrated into the *P. pastoris* genome by a homologous recombination using the AOXI promoter. Successful transformants with a high copy number were isolated by Zeocin selection (1000 μg/mL) [36].

### 2.3. Protein Expression, Purification, and Basic Measurements

A selected *Pichia pastoris* GS115/pPICZB-Cecytb-2-H_6_ strain was grown for 24 h in 25 mL BMGY medium containing Zeocin (25 μg/mL) at 30 °C. Then, the cultured cells were directly transferred to 250 mL BMGY medium and grown for 48 h at 30 °C. Then, the cells were harvested by centrifugation and re-suspended in 250 mL BMMY medium containing 2% methanol and 40 μg/mL of L-histidine monohydrochloride for the induction of protein expression. The medium was supplemented with 5 mL of methanol every 12 h and with 40 μg/mL of L-histidine monohydrochloride every 24 h. After 96 h of induction, the cells were harvested by centrifugation and stored at −80 °C until use.

The cell pellet was suspended in 50 mM potassium phosphate buffer (pH 7.0) containing 2 M sorbitol, 0.1 mM EDTA, and 0.1 M DTT for lysis. After adding 1 mg zymolyase-100T (Nacalai Tesque, Kyoto, Japan) to 15 g (wet weight) of the pellet, the suspension was incubated at 35 °C for 16 h with gentle shaking. The cell spheroplasts were harvested by centrifugation at 40,000× *g*, 4 °C, for 15 min and re-suspended in 50 mM potassium phosphate buffer (pH 7.0) containing 0.65 M sorbitol, 0.1 mM EDTA, 0.1 mM DTT, and 1 mM PMSF. The mixture was sonicated (20 min total; 0.7 pulse, 2 s intervals, output level 9.0; Astrason 3000, MISONIX, Qsonica, Newtown, CT, USA) on ice and then centrifuged at 40,000× *g*, 4 °C, for 15 min. The supernatant was then ultra-centrifuged at 100,000× *g*, 4 °C, for 60 min to obtain a microsomal pellet. The microsomal pellet was suspended and homogenized in 50 mM potassium phosphate buffer (pH 7.4) containing 10% (*v*/*v*) glycerol. The homogenate was frozen in liquid nitrogen and stored at −80 °C until use.

The concentration of the Cecytb-2-H_6_ protein in the microsomal homogenate was determined by UV–visible redox difference spectrum using a difference extinction coefficient of *ε_561–575_* = 27.7 mM^−1^cm^−1^, a value determined for adrenal cytochrome *b*_561_ [1]. The redox difference spectra for the AsA-reduced *minus* air-oxidized and/or dithionite-reduced *minus* air-oxidized were used for the determinations. The microsomal homogenate was solubilized by the addition of solid octyl-β-D-glucoside (OG), in which the homogenate containing 10 g (wet weight) of the microsomal pellet was solubilized by 2 g of OG. The solubilization buffer (50 mM potassium phosphate buffer, pH 7.4, containing 10% (*v*/*v*) glycerol, 1% (*w*/*v*) OG, 300 mM NaCl, 20 mM imidazole; buffer A) was added to the microsome homogenate, and the mixture was stirred gently on ice for 1 h. After ultra-centrifugation at 100,000× *g*, 4 °C, for 30 min, the supernatant was collected and applied to a His GraviTrap column (GE Healthcare Japan, Hino, Japan) equilibrated with buffer A. The column was then washed with the same buffer but containing 20–80 mM imidazole. The adsorbed Cecytb-2-H_6_ protein was eluted with 50 mM potassium phosphate buffer (pH 7.4) containing 10% (*v*/*v*) glycerol, 1% (*w*/*v*) OG, 300 mM NaCl, and 500 mM imidazole. The eluted fractions were desalted by gel filtration through a PD-10 column (GE Healthcare Japan, Hino, Japan) equilibrated with 50 mM potassium phosphate buffer (pH 7.4) containing 10% (*v*/*v*) glycerol, 1% (*w*/*v*) OG, and 300 mM NaCl. If necessary, the purified protein was concentrated using a centrifugal concentrator (MWCO = 10,000; Millipore, Merck Ltd., Tokyo, Japan), frozen in liquid nitrogen, and stored at −80 °C until use. SDS-PAGE analysis was conducted using 4% polyacrylamide as the stacking gel and 12% polyacrylamide as the separating gel. The sample for SDS PAGE was prepared by mixing an aliquot of the protein with an equal volume of sample buffer (10 mM Tris-HCl buffer (pH 6.8) containing 1% (*w*/*v*) SDS, 20% (*w*/*v*) sucrose, 1% (*v*/*v*) 2-mercaptoethanol). The protein content was determined by the modified Lowry method [37], using bovine serum albumin as the standard. On the other hand, the concentration of the Cecytb-2-H_6_ protein was determined spectrophotometrically in the dithionite-reduced state using an extinction coefficient of 39.5 mM^−1^cm^−1^ at 561 nm. The heme *b* content of the purified Cecytb-2-H_6_ protein was determined using the pyridine hemochrome method, as previously described [1,38].

### 2.4. Diethylpyrocarbonate (DEPC) Treatment of Purified Cecytb-2-H_6_

The oxidized form of Cecytb-2-H_6_ (1.4 μM) in 50 mM K-Pi buffer (pH 7.4) containing 0.1% n-dodecyl-β-D-maltoside (DDM) and 10% glycerol was treated with DEPC (final 0.5 mM by addition of 30 mM of DEPC in ethanol) at room temperature for 30 min while recording the spectral changes in the 700–200 nm region in difference spectral mode. After the treatment, the sample was desalted by column chromatography to remove unreacted DEPC. The DEPC-treated sample, when analyzed, was compared alongside a control and an ethanol-treated sample.

### 2.5. EPR Measurement

A concentrated purified Cecytb-2-H_6_ protein sample (≈60 μM in 50 mM potassium phosphate buffer, pH 7.4, containing 10% (*v*/*v*) glycerol and 1% (*w*/*v*) OG) in an air-oxidized state was placed inside a quartz EPR sample tube and frozen in liquid nitrogen. X-band EPR measurements were carried out using a Varian E-109 EPR spectrometer, as previously described [39]. The instrumental parameters were as follows: microwave power, 10 mW; modulation frequency, 100 kHz; and modulation amplitude, 1 mT. An Oxford flow cryostat (ESR-900) was used for the cryogenic temperatures.

### 2.6. Redox Titration

Spectroscopic redox titrations were performed essentially as described by Takeuchi et al. [40] using a Shimadzu UV-2400PC spectrometer equipped with a thermostatted cell holder connected to a low-temperature thermo bath (RC-6CS, LAUDA Scientific GmbH, Lauda-Königshofen, Germany). A custom anaerobic cuvette (1-cm light path, 5 mL sample volume) equipped with a combined platinum and Ag/AgCl ORP electrode (9300-10D, HORIBA Ltd., Kyoto, Japan) and a screw-capped side arm was used. A purified Cecytb-2-H_6_ sample (final, ≈5 μM) in 50 mM potassium phosphate buffer (pH 7.4), 0.1% (*w*/*v*) DDM, and 10% (*v*/*v*) glycerol was mixed with redox mediators (potassium ferricyanide, 60 μM; quinhydrone, 20 μM; 1,2-naphthoquinone, 20 μM; phenazine methosulfate, 20 μM; duroquinone, 40 μM; 2-hydroxy-1,4-naphtoquinone, 5 μM; riboflavin, 20 μM). Then, the sample solution was placed in a cuvette after passing through a cellulose acetate filter (DISMIC-25cs, 0.45 μm; Toyo Roshi Kaisha, Ltd., Tokyo, Japan). The sample was kept under a flow of moistened argon gas to exclude dioxygen and was continuously stirred with a small magnetic stirrer (CC-301, SCINICS, Tokyo, Japan) inside. Reductive titration was performed at 20 °C by the addition of small aliquots of a sodium dithionite (5 or 20 mM) solution through a needle in the rubber septum on a side arm. For a subsequent oxidative titration, potassium ferricyanide (5 or 20 mM) was used as the titrant. In an appropriate interval, visible absorption spectra and redox potentials were recorded. The reported potentials were referenced to a standard hydrogen electrode (SHE). The changes in absorbance (A_561.0_ *minus* A_567.0_, the isosbestic point of the Cecytb-2-H_6_ protein) were corrected considering dilution effects and were converted to heme reduction levels (%). Then, the datasets were analyzed with Igor Pro (v. 6.37) by employing a Nernst equation with a single redox component,f(x) = base + (max × 1/(1 + exp((x1 − x)/rate1)))(1)
or by a Nernst equation with two redox components,f(x) = base + (max × 1/(1 + exp((x1 − x)/rate1))) + (max × 1/(1 + exp((x2 − x)/rate2)))(2)
where f(x) is the percentage of the heme reduction level and x is the solution reduction potential, with x1 and x2 being the midpoint potentials of interest. For the other parameters, rate1 and rate2 were both fixed as 25.262 (mV), and base and max were set to 100 and −50, respectively. During the analysis, we did not consider the coupling of the two heme centers in their redox potentials for simplicity [41]; these two heme centers are expected to be rather distant from each other in the protein molecule [24] and, therefore, would not cause significant conformational changes during the electron transfer reactions.

### 2.7. Stopped-Flow Analysis

The electron transfer reactions of the purified Cecytb-2-H_6_ protein with AsA were measured by a stopped-flow method using an RSP-100-03DR stopped-flow rapid-scan spectrometer (UNISOKU Co. Ltd., Hirakata, Japan), as previously described [33,42,43]. Three kinds of buffers with different pH values (50 mM potassium phosphate buffer for pH 6.0, 7.0, and 7.4, and 50 mM sodium acetate buffer for pH 5.0, with each containing 1% (*w*/*v*) OG) were used to measure pH dependency. Upon the analyses, different concentrations of the Cecytb-2 protein (2, 3, 4 μM) and AsA (2, 4, 8, 16 mM) were mixed in a 1:1 (*v*/*v*) ratio at room temperature (≈25 °C). The heme reduction of the Cecytb-2-H_6_ protein was followed spectrophotometrically by the absorbance change at 426 or 427 nm. The time-dependent absorbance changes were fitted to an equation with a linear combination of three exponential functions, as previously described [44]. Then, three apparent rate constants were evaluated based on their dependencies on the AsA concentration and protein concentration.

### 2.8. Pulse Radiolysis

Pulse radiolysis experiments were performed using an electronic linear accelerator at the Institute of Scientific and Industrial Research, Osaka University, as previously described [33,45]. The purified Cecytb-2-H_6_ protein in 10 mM potassium phosphate buffer, pH 7.0, with 1% (*w*/*v*) OG and 10 mM AsA was placed inside a quartz cell and then bubbled with N_2_O gas for about 2 min. The concentration of the pulse-generated MDA radical was measured at 360 nm using a molar extinction coefficient of 3300 M^−1^cm^−1^. In our present study, the MDA radical concentration was maintained to be much smaller than the Cecytb-2 protein concentration. Oxidation and reduction of the Cecytb-2-H_6_ protein after the radiation pulse were monitored by following absorbance changes at 435 nm and/or various wavelengths in the Soret region using a spectrophotometer.

### 2.9. Protein Digestion and MALDI-TOF-MS and Online HPLC-Orbitrap MS/MS Measurements

DEPC-treated or untreated Cecytb-2-H_6_ protein samples were concentrated to about 50 μM. Then, the samples (10 μL) were each treated with either 0.5 μL of TPCK-treated trypsin, *Staphylococcus aureus* V8 protease, or α-chymotrypsin (for each protease, 1 mg/mL in 20 mM K-Pi buffer pH 7.0) at room temperature for 24 h. For the MALDI-TOF-MS measurements, Ultraflex III-HID (Bruker) (at ISIR, Osaka University, Ibaraki, Japan) was used after mixing the digested sample (1 μL) with a matrix solution (1 μL), followed by direct analysis by depositing on the target plate (dried droplet method). For the online HPLC MS/MS measurements, LTQ Orbitrap Discovery (Thermo-Fischer Scientific Japan, Tokyo, Japan) connected with Pradigm MS2 (Michrom BioResources, Auburn, CA, USA) and a Nano HPLC capillary column (0.1 × 150 mm, 3 μm) (at the Research Facility Center for Science and Technology, Kobe University, Kobe, Japan) was used. For the online HPLC separation, two gradient conditions (45 min and 90 min) of mobile phase A (CH_3_CN/H_2_O = 2/98, 0.1% HCOOH) and mobile phase B (CH_3_CN/H_2_O = 90/10, 0.1% HCOOH) for reverse-phase separation were used, and the eluates from the HPLC capillary column were directly introduced into LTQ Orbitrap and analyzed (ionization, Nano ESI Positive mode; source voltage, 2.5 kV; capillary temperature, 200 °C; tube lens voltage, 115 V; mass range, 300–1800 m/z). MASCOT analyses were conducted (type of search, MS/MS Ion Search; variable modification condition, Oxidation (M), Carbethoxy (CHKSTY), Carbethoxy (Protein N-term)); max missed cleavages, 9.

### 2.10. Structural Studies of the Cecytb-2 Protein

The homology model of the Cecytb-2 protein was generated using HOMCOS (HOMology modeling of COmplex Structure) (http://homcos.pdbj.org), a server for modeling complex 3D structures based on 3D molecular similarities derived from template complex 3D structures in the PDB. For a given amino acid sequence or a docked chemical structure (i.e., heme and AsA), the server provides a list of contacting molecules in the PDB and the predicted complex 3D structure based on the template PDB structures, in which BLAST is employed for the amino acid sequence search. Then, a sequence-replaced 3D model structure was generated. Then, using USCF-Chimera (v. 1.19) and Modeller (v. 10.7) software, homology models were generated.

The homology model of the Cecytb-2 protein was also generated from the AlphaFold Protein Structure Database (AlphaFold DB (July 2022 release), https://alphafold.ebi.ac.uk), an AI system developed by Google DeepMind that predicts a protein’s 3D structure from its amino acid sequence. The generated structure was curated and provided at this website: https://alphafold.ebi.ac.uk/entry/O16271.

### 2.11. Ferrozine-Based Colorimetric Assay for Transmembrane Electron Transfer Activity

Proteoliposomes containing the purified Cecytb-2-H_6_ protein in the membranes and AsA (final concentration, 100 mM) were entrapped inside the lumen, as previously described [46,47], with minor modifications. Briefly, 24 mg of Presome TSU (DPPC–cholesterol = 95:5) was dissolved in 900 μL of buffer NA (50 mM sodium phosphate buffer (pH 7.0) containing 150 mM NaCl, 100 mM AsA) by vortexing and sonication for 3 min. Then, 100 μL of the purified Cecytb-2 protein (12 μM), dissolved in 50 mM potassium phosphate buffer (pH 7.4) containing 10% (*v*/*v*) glycerol, 1% (*w*/*v*) OG, and 300 mM NaCl, was added to the Presome TSU solutions. The detergent was removed by dialysis at 4 °C for 16 h. After the dialysis, the solutions were treated with 3 cycles of freezing and thawing using liquid nitrogen. To obtain the single-lamellar vesicles, the resulting proteoliposome mixture was extruded through a polycarbonate filter (pore size, 500 nm; Nuclepore, Pleasanton, CA, USA) using a mini-extruder (LiposoFast Basic, Avestin, Ottawa, Canada) 29 times.

The proteoliposomes were finally gel-filtered twice through an Ampure SA column against 50 mM potassium phosphate buffer (pH 7.0) containing 150 mM NaCl. Ferric ions in the form of FeCl_3_ (final, 200 μM) and PTDS (3-(2-pyridyl)-5,6-*bis*(4-sulfophenyl)-1,2,4-triazine) (i.e., ferrozine, final, 100 μM) [48] were added to the outside of the proteoliposomes in the same buffer to produce a time-dependent build-up of the purple color due to the formation of a ferrous–ferrozine complex produced by transmembrane electron transfer via the reconstituted Cecytb-2-H_6_ protein. The color change, which peaked at 562 nm, was monitored by a Shimadzu UV-2400 PC spectrophotometer. In the final stage of each experiment, the integrity of the proteoliposomes was confirmed by the addition of Triton X-100 (final 1%) and the subsequent build-up of the peak at 562 nm. For a control experiment, 100 μL of 50 mM potassium phosphate buffer (pH 7.4) containing 10% (*v*/*v*) glycerol, 1% (*w*/*v*) OG, and 300 mM NaCl was used instead of the Cecytb-2-H_6_ protein solution during the reconstitution.

### 2.12. Production and Purification of Anti-Cecytb-2C-Terminal Peptide Antibodies

For the preparation of an antigen to produce site-specific antibodies, a coding region for the C-terminal end (PVPWRREKTPDELK) of the Cecytb-2 protein was amplified by PCR using the forward primer (TTAAGATCTCGGTAGTCCAGTGCCGTGGAGAAG, containing *Bgl*II site) and reverse primer (GCGGCTGCAGTTACTTTAGCTCATCTGG, containing *Pst*I site). The PCR product was digested and introduced into the *Bgl*II/*Pst*I site of a pQE41 vector (QIAGEN K.K., Hino, Japan) for the expression of a fusion protein, i.e., dihydrofolate reductase (DHFR) fused with Cecytb-2 C-terminus peptide on its C-terminus and a 6xHis-tag sequence on its N-terminus (Appendix A) [33]. The resultant plasmid, pQE41-DHFR-Cecytb-2C-terminus, was transformed into *E. coli* XL1Blue competent cells, and their DNA sequence was confirmed. The transformants were grown in LB medium containing ampicillin (100 μg/mL) at 37 °C until A_600_ reached 0.6. Then, IPTPG was added (final, 1 mM), and cultivation was continued for 5 h. The cells were harvested by centrifugation and were stored at −80 °C until use. The fusion protein was purified using a His-GraviTrap column, a method similar to that employed for the purification of the Cecytb-2-H_6_ protein, but without the addition of OG. The immunization of a rabbit with the purified DHFR-Cecytb-2C-terminal peptide was conducted by Takara Bio Inc. (Kusatsu, Japan). The IgG fraction antibody of the serum was purified by ammonium sulfate fractionation (final, 1.5 M ammonium sulfate in 50 mM potassium phosphate buffer, pH 7.0, twice). The partially purified IgG fraction was applied onto the DHFR-conjugated NHS column (GE Healthcare Japan, Hino, Japan), and the eluted fraction was used as the purified anti-Cecytb-2 IgG.

### 2.13. Western Blot Analysis

Cultivation of *C. elegans* in liquid medium was performed as follows. N2 worms were suspended in M9 buffer, which consists of 22 mM KH_2_PO_4_, 22 mM Na_2_HPO_4_, 85 mM NaCl, and 1 mM MgSO_4_. Subsequently, this suspension was added to 300 mL of NGM liquid medium containing 0.25% (*w*/*v*) Bacto Peptone, 0.3% (*w*/*v*) NaCl, 5 μg/mL cholesterol, 25 mM potassium phosphate buffer pH 6.0, 1 mM MgSO_4_, and 1 mM CaCl_2_. Additionally, *E. coli* OP50 pellets from a 250 mL overnight culture were incorporated into the medium. Following three days of cultivation at 20 °C, a pellet of the new *E. coli* OP50 from a 250 mL overnight culture was added, and the mixture was incubated for an additional day at 20 °C. To isolate the worm pellet, an equal volume of 60% (*w*/*v*) sucrose was added to the culture medium, and the mixture was then centrifuged at 4000× *g* for 2 min. The worm pellet was suspended in 50 mM potassium phosphate buffer (pH 7.0) containing 0.65 M sorbitol, 0.1 mM EDTA, 0.1 mM DTT, and 1 mM PMSF. The mixture was sonicated (10 min total; 0.7 pulse, 2 s intervals, output level 5.0) on ice and then centrifuged at 30,000× *g*, 4 °C, for 15 min. The supernatant was ultra-centrifuged at 100,000× *g*, 4 °C for 60 min to obtain a microsomal pellet. The microsomal pellet was suspended and homogenized in 50 mM potassium phosphate buffer (pH 7.4) containing 10% (*v*/*v*) glycerol. The supernatant fraction containing soluble proteins and the membrane fraction were collected, and each was frozen in liquid nitrogen and stored at −80 °C until use. These samples were each treated with SDS and were separated by SDS-PAGE with a 14% polyacrylamide gel. Protein bands in the gel were transferred electrophoretically onto a nitrocellulose membrane in 20 mM glycine, 25 mM Tris, and 20% (*v*/*v*) methanol. The membrane was first stained with 0.1% (*w*/*v*) Ponceau S in 1% (*v*/*v*) acetate to check the transferred proteins. The membrane was then de-stained by washing with distilled water and blocked with 1% (*w*/*v*) bovine serum albumin in PBS-T buffer. The anti-Cecytb-2 rabbit antibody was used as the primary antibody, and the anti-rabbit IgG horseradish peroxidase-conjugated goat antibody (Wako) was used as the secondary antibody. To detect the endogenous Cecytb-2 protein on the gel, 4-chloro-1-naphthol and hydrogen peroxide were used as substrates [33,49].

### 2.14. In Situ Hybridization Analysis

The DIG-labeled anti-Cecytb-2 RNA probe was synthesized in vitro by using T3 RNA polymerase (Promega K.K., Tokyo, Japan) and a DIG RNA Labeling Kit (Roche Diagnostics K.K., Tokyo, Japan) with pBlueScript II KS (+)/Cecytb-2 as a template. All procedures followed the recommended protocols. In situ hybridization analysis for Cecytb-2 mRNA was conducted as previously described [50] with some modifications. Briefly, worms were harvested by centrifugation, washed three times with M9 buffer, and then fixed by immersion in 4% (*w*/*v*) paraformaldehyde in 0.1 M potassium phosphate buffer (pH 7.2) at 4 °C overnight. Then, the worms were washed 3 times in PBS-T buffer, permeabilized by treatment with proteinase K, followed by washing 3 times in PBS-T buffer, twice in 0.1 M triethanolamine (pH 8.0), once in 0.1 M triethanolamine and 0.5% (*v*/*v*) acetic anhydride (pH 8.0), and 3 times in PBS-T buffer. The DIG haptens were detected by the primary antibody (1:100 diluted anti-DIG rabbit IgG, Invitrogen) containing 2% (*w*/*v*) BlockAce (Sumitomo Pharma Promo Co. Ltd., Suita, Japan) and 1% BSA. Then, the bound primary antibody was detected by 1:200 diluted Alexa Fluor 488-conjugated anti-rabbit IgG goat IgG antibodies (Invitrogen, Thermo-Fisher Scientific Japan, Tokyo, Japan). The images were observed under a light fluorescence microscope (BZ-9000, KEYENCE Corporation, Osaka, Japan) with excitation at 488 nm and emission at 530 nm.

### 2.15. Immunostaining

Harvested worms were fixed with 3% (*w*/*v*) paraformaldehyde in 0.1 M potassium phosphate buffer (pH 7.2) containing 25% methanol at 4 °C for 1 h. The fixed worms were washed with TTB (100 mM Tris-HCl buffer (pH 7.4) containing 1% (*v*/*v*) Triton X-100) 3 times. The washed worms were treated in TTB containing 1% (*v*/*v*) 2-mercaptoethanol at 37 °C for 1 h, followed by washing in 10 mM NaBO_3_ (pH 9.2) 3 times. Then, the worms were treated in 10 mM NaBO_3_ (pH 9.2) containing 0.3% (*v*/*v*) hydrogen peroxide for 1 h at room temperature, followed by washing in 10 mM NaBO_3_ (pH 9.2) 3 times. The primary antibody used was anti-Cecytb-2 rabbit IgG, which was diluted 1:100 with 4% BlockAce in PBS-T solution. The secondary antibody, Alexa Fluor 488-conjugated anti-rabbit IgG goat IgG, was used at a 1:200 dilution. The fluorescent signals were detected under a light fluorescence microscope (BX51, OLYMPUS, Tokyo, Japan) with excitation at 488 nm and emission at 530 nm.

### 2.16. RNAi

RNAi experiments were performed following a method previously described with some modifications [51]. Briefly, the Cecytb-2 gene in the pBlueScript II KS (+) vector was PCR-amplified and then digested using the same method employed for cloning. The PCR product was introduced into the L4440 feeding RNAi plasmid. Bristol N2 worms were grown on NGM plates containing 1 mM IPTG and 50 μg/mL ampicillin and were then fed with *E. coli* HT115 (DE3) transformed with the L4440 RNAi plasmid. An empty L4440 plasmid was used as a control. To maintain the ferrous-deficient conditions, the NGM plates were supplemented with 20, 40, or 60 μM of 2,2′-dipyridyl (BP). The RNAi-treated L4 worms were each transferred to a new NGM plate supplemented with BP and were cultured. Then, the RNAi-treated adult worms were each transferred to a new NGM plate supplemented with BP and allowed to lay eggs. Then, the adult worms were removed, and the hatched eggs were analyzed using various tests.

## 3. Results

### 3.1. Seven Cytochrome b_561_ Homologues in C. elegans

BLAST searches using the amino acid sequences of human adrenal cytochrome *b*_561_ (CGcytb) and human SDR-2 as queries were performed using the Wormbase server (http://www.wormbase.org/tools/blast_blat). Extensive searches showed seven cytochrome *b*_561_-like genes in the genome of *C. elegans*. We named them Cecytb-1 to Cecytb-7 (for the genes *F55H2.5*, *F39G3.5*, *F39G3.4*, *C05D12.1*, *M03A1.3*, *M03A1.8*, and *C13B4.1*, respectively). Alignments of the deduced amino acid sequences of these homologs are shown in Appendix A. The Cecytb-1 protein had the highest homology (37%), whereas the Cecytb-2 protein showed a lower homology (26%), to human CGcytb and Dcytb. Therefore, in our previous study [5], we grouped Cecytb-1 as a member of the A subfamily, but Cecytb-2 was not classified in any subfamilies due to its extensive deviation. However, our recent study showed that the distribution of the Cecytb-1 protein in adult worms was not consistent with the expected participation in neural functions (Unpublished results, available upon request from the corresponding author). Therefore, we shifted our focus to the Cecytb-2 protein, with the expectation that it would have similar roles to those of CGcytb and Dcytb. The Cecytb-2 protein has an expected sequence of 251 amino acid residues with a molecular mass of 28,127 Da (Appendix A). The hydropathy plot analysis confirmed a six-transmembrane α-helix structure with two pairs of conserved His residues (His67, His101, His135, and His174) for the possible axial ligands of two *b*-type heme prosthetic groups residing on both sides of the membranes, which are common properties of members of the family (Appendix A) [5,52].

### 3.2. Biochemical and Biophysical Properties of Cecytb-2

To characterize the biochemical and biophysical nature of the Cecytb-2 protein, we constructed its heterologous expression system using *Pichia pastoris* cells by adding a linker sequence followed by a 6xHis-tag sequence at the C-terminus of the native Cecytb-2 protein (Appendix A). The microsomal fractions obtained from the transformed *P. pastoris* cells upon induction with methanol showed a characteristic visible redox difference spectrum (AsA-reduced) with a peak at 560 nm, indicating the successful expression of the Cecytb-2-H_6_ protein in an intact holo-form (Figure 1A). The purified Cecytb-2-H_6_ protein showed characteristic visible absorption spectra with peaks at 412 nm for the air-oxidized form and at 426, 529, and 560.5 nm for the dithionite-reduced form, as a member of the cytochrome *b*_561_ family (Figure 1B). The purified protein could be reduced with AsA very efficiently (Figure 1B). SDS-PAGE analysis of the purified Cecytb-2-H_6_ protein showed a single protein band at 29.2 kDa (Appendix A), very close to its theoretical value (29.108 kDa). Pyridine hemochrome analysis indicated that the purified protein contained 1.69 (±0.18 SD) molecules of *b*-type heme per 1 protein molecule (Appendix A), being consistent with the view that a member of the cytochrome *b*_561_ family contains two heme *b* prosthetic groups per molecule.

The EPR spectra of the purified Cecytb-2-H_6_ protein in an air-oxidized state exhibited signals at g_z_ = 3.27 and g_y_ = 2.26 when measured at 15 K and at g_z_ = 3.65 when measured at 8 and 5 K (Figure 2). The EPR signal of g = 2.94, which was occasionally observed in a denatured state [23,44,53,54], was not observed in our preparation. The g_z_ = 3.65 signal was assigned as the highly anisotropic low-spin (HALS) species, whereas the g_z_ = 3.27 and g_y_ = 2.26 signals belong to a usual rhombic EPR species. These EPR parameters were more similar to those of human Dcytb [23] than those of CGcytb or 101F6 [39,41,44,55,56,57,58], although the g_z_ = 3.65 signal was somewhat broader than other HALS-type g_z_ signals observed in the cytochrome *b*_561_ family so far, as shown in Appendix A.

Redox potential measurements by monitoring changes in α-band absorbance were conducted for the purified Cecytb-2-H_6_ protein. The results for the reductive phase are shown in Figure 3A. Analysis by fitting using a Nernst equation with a single exponential function indicated that the midpoint potential was +96.3 (±1.6) mV, although the fitting was not satisfactory (Figure 3B). Analysis by fitting with a Nernst equation featuring two exponential functions yielded a very satisfactory fit with two midpoint potentials at +126.2 (± 1.6) mV and +66.9 (± 1.6) mV (Figure 3C). These results showed a close resemblance to those of human Dcytb measured by a potentiometric titration using a MOTTLE cell [23]. Oakhill et al. suggested that the two hemes of human Dcytb may have very close midpoint potentials or different potentials by up to 60 mV [23]. In our case, however, fitting the data to a Nernst equation with two components yielded a much better result.

### 3.3. Stopped-Flow Studies of the Reaction of Cecytb-2 with AsA

To analyze the heme reduction process of the Cecytb-2-H_6_ protein with AsA, the purified Cecytb-2-H_6_ protein in the oxidized state and AsA were mixed in a stopped-flow apparatus, and the following absorption changes at 426 or 427 nm were recorded. First, concentration dependencies of the heme reduction process on the AsA concentration (Figure 4A) and Cecytb-2-H_6_ protein concentration (Appendix A) were examined at pH 7.0 and 7.4, respectively. When the absorbance changes at 426 nm were fitted with a linear combination of three exponential functions, we obtained a satisfactory fit for all the concentrations examined (Figure 4A, Appendix A), as previously accomplished for Dcytb [44]. It was reported previously that, for the adequate fitting of the heme reduction process of adrenal CGcytb with AsA, at least four exponential functions were required [42]. These results indicated that similar electron transfer mechanisms are shared between Cecytb-2-H_6_ and human Dcytb. From the AsA concentration dependency of *k*_app1_, the second-order rate constant (*k*_1_) for the heme reduction with AsA was estimated as 3.87 × 10^3^ (M^−1^s^−1^) (Figure 4B). Then, we examined the heme reduction process of Cecytb-2-H_6_ with AsA at three different pH values. The fastest heme reduction occurred at pH 7.0. However, interestingly, at pH 5.0, there was no initial lag time for the heme reduction process, although the amplitude of the absorbance change became smaller (Figure 4C). We previously observed a significant initial lag time at pH 5.0 for adrenal CGcytb [42] and plant cytochrome *b*_561_ [33]. Fitting analyses using a linear combination of three exponential functions on these kinetic data confirmed that all three *k*_app_ values did not show any appreciable pH dependency, and, particularly, *k*_app1_ at pH 5.0 showed a high value comparable to those at pH 6.0 or pH 7.0 (Appendix A). Cecytb-2-H_6_ protein concentration dependency on the heme reduction process with AsA was then examined at three different concentrations (Appendix A). As expected, there was no appreciable concentration-dependent change in each of the three *k*_app_ values (Appendix A).

### 3.4. Inhibition of Electron Transfer from AsA by DEPC Treatment of Cecytb-2

In the next step, we examined the effects of DEPC treatment on the Cecytb-2-H_6_ protein’s electron-accepting ability from AsA. Our previous studies showed that the electron transfer activity from AsA to oxidized heme was significantly inhibited upon DEPC-treatment for CGcytb [40] and for plant Zmcytb561 [34], but not at all for 101F6 [39]. We previously explained that the cause of the inhibition was the specific *N*-carbethoxylation of one of the His residues of the heme axial ligand (corresponding to His101 of Cecytb-2) and the well-conserved Lys residue (corresponding to Lys98 of Cecytb-2) near the cytosolic heme center (Appendix A) [34,40]. As expected, the DEPC treatment of the Cecytb-2-H_6_ protein caused a significant inhibition of the electron acceptance ability from AsA, both in the initial rate (*k*_1_) (Appendix A) and in the final reduction level (Appendix A). However, their extents of inhibition were slightly lower than those observed for CGcytb [40] and for plant Zmcytb561 [34]. To precisely identify the DEPC modification sites of the Cecytb-2-H_6_ protein, we conducted MALDI-TOF analyses on the peptides produced by digestion with trypsin, V8-protease, and α-chymotrypsin. The analyses indicated that multiple *N*-carbethoxylations, including those at His101 and Lys98, occurred as previously reported [40]. In a subsequent step, using online HPLC MS/MS ion search analysis in MASCOT, we confirmed that both Lys98 and His101 were indeed highly *N*-carbethoxylated (Appendix A–C). The MS/MS analysis indicated that, in addition to Lys98 and His101, Ser2, Ser3, His24, Lys50, Lys52, Lys57, Lys61, Tyr89, Lys94, Cys204, Lys245, and Thr246 were also carbethoxylated (Appendix A). Among them, Ser2, Ser3, Tyr89, Lys94, Lys98, His101, Lys245, and Thr246 faced the cytosolic side and only Lys94, Lys98, and His101 were highly conserved (Appendix A). Accordingly, it might be very reasonable to consider that the *N*-carbethoylations at Lys98 (and possibly at Lys94) (highly conserved on the cytosolic side) and at His101 (fully conserved as a heme axial ligand on the cytosolic side) (Appendix A) would cause the inhibition of electron acceptance from AsA on the cytosolic side in a very similar molecular mechanism as proposed previously for other cytochrome *b*_561_ protein family members [59]. Our present measurements of the midpoint potentials for the two heme centers of the DEPC-treated Cecytb-2-H_6_ protein indicated that the heme center with a lower redox potential (heme *b*_L_) was influenced more significantly than the other heme (heme *b*_H_), showing a further lowering in its midpoint potential (from +66 to +45 mV) (Figure 3D). This observation was in accordance with our previous results for DEPC-treated CGcytb (bovine), which indicated that only heme *b*_L_, possibly located on the cytosolic side (i.e., extravesicular side), was affected [40].

### 3.5. Pulse Radiolysis Study of Cecytb-2

To confirm whether the Cecytb-2 protein can react with the MDA radical, as observed for other cytochromes *b*_561_ [18,33,45], we conducted pulse radiolysis experiments. The purified Cecytb-2-H_6_ protein was diluted in a buffer containing 10 mM AsA and 1% OG, and then, a pulse beam was irradiated to the sample solution to generate MDA radicals [45]. The absorbance change at 435 nm was measured to analyze the electron transfer reactions of the Cecytb-2 protein. In our experimental condition, the Cecytb-2-H_6_ protein was in an almost fully reduced state (≈85%) in the presence of AsA (10 mM) (Figure 5). Therefore, the decrease in the time range of milliseconds in absorbance at 435 nm upon pulse irradiation corresponds to the heme oxidation of Cecytb-2-H_6_ by the generated MDA radical (Figure 5A). Dependency of the apparent rate constant for the oxidation reaction (*k*_1app_) on the Cecytb-2-H_6_ protein concentration showed that the second-order rate constant (*k*_1_) was 3.1 × 10^6^ (M^−1^s^−1^) (Figure 5C), which is much slower than those of other cytochrome *b*_561_ family proteins (5.0 × 10^7^ M^−1^s^−1^ for h101F6 [18]; 2.6 × 10^7^ M^−1^s^−1^ for bovine adrenal CGcytb [45]; 1.0 ×10^7^ M^−1^s^−1^ for *Zea mays* cytochrome *b*_561_ [33]). After the initial rapid oxidation of the reduced heme, the decreased absorbance at 435 nm recovered to the initial level (Figure 5B). This slower process corresponds to the re-reduction of the oxidized heme center by AsA in the surrounding medium. The apparent rate constants of the re-reduction reaction (*k*_2app_) were found to be almost independent of the protein concentration, with a mean value of 0.38 (s^−1^) (Figure 5D). These kinetic experiments indicated that the Cecytb-2-H_6_ protein could accept electrons from AsA very rapidly and donate them to MDA radicals, but with a much lower reactivity than other cytochrome *b*_561_ family members.

Thus, our present results on EPR, heme redox potentials, and reaction kinetics with AsA indicate that the Cecytb-2 protein is more similar to human Dcytb than CGcytb, suggesting the possibility that the Cecytb-2 protein functions as an AsA-specific transmembrane ferric reductase in *C. elegans*. The lowered reactivity toward the MDA radical seems consistent with this view. Indeed, we recently reported that (chemically produced) a reduced form of purified Cecytb-2 protein acted as a ferric reductase, and this activity became enhanced upon reconstitution into phospholipid bilayer nanodisc [31].

### 3.6. AsA-Dependent Transmembrane Ferric Reductase Activity of Cecytb-2

Accordingly, in the next step, a ferrozine-based colorimetric assay was conducted to clarify the transmembrane ferric reductase activity of the Cecytb-2 protein. For the assay, proteoliposomes in which the purified Cecytb-2-H_6_ protein was reconstituted into the membranes, whereas AsA was entrapped inside the lumen, were prepared using the detergent dialysis and extrusion method [46,47]. The proteoliposomes were found to be very stable, as previously shown for similar proteoliposomes containing bovine CGcytb instead [46]. The addition of ferric ions and PDTS (3-(2-pyridyl)-5,6-*bis*(4-sulfophenyl)-1,2,4-triazine) (i.e., ferrozine) to the outside of the proteoliposomes caused a time-dependent build-up of the purple color due to the formation of a ferrous–ferrozine complex, which peaked at 562 nm, as shown in Figure 6A. On the other hand, the liposome without the purified Cecytb-2-H_6_ protein but containing AsA inside the vesicles showed a very shallow increase in absorbance at 562 nm, probably reflecting a very slow leakage of the entrapped AsA (Figure 6B). Thus, our colorimetric assay confirmed that the Cecytb-2-H_6_ protein exhibits transmembrane ferric reductase activity utilizing AsA as the sole source of electrons.

### 3.7. Structural Study of the Cecytb-2 Protein

The putative structural models of the Cecytb-2 protein were calculated by employing two approaches (HOMCOS and AlphaFold). HOMOS identified two existing X-ray crystal structures (i.e., *Arabidopsis thaliana* CYB561B2 (4O7G.pdb, 4O6Y.pdb, and 4O79.pdb) [60] and human Dcytb (CYBR1_HUMAN) (5ZLE.pdb and 5ZLG.pdb) [24] as suitable 3D templates, as expected. The alignments of the two sequences provided by HOMOS agreed well with the predicted transmembrane segments. For the AlphaFold approach, the predicted Cecytb-2 protein structures were very similar to those from the HOMOS approach. These predicted structures highly overlap with the two existing X-ray crystal structures. The two docked hemes and AsA ligands (and a Zn^2+^ ion shown in the Dcytb (CYBR1_HUMAN) structure) were included in the modeling (Appendix A). The obtained Cecytb-2 protein surface shows similar troughs to those in the X-ray crystal structures, serving as potential binding sites for AsA and ferric substrates on both sides (Appendix A). The overall pattern of the surface charge around the AsA-binding sites on both sides of the protein is rather similar in the two existing X-ray crystal structures (Appendix A) [24].

Thus, the calculated homology models of the Cecytb-2 protein were consistent with our various biophysical and biochemical observations in the present study (EPR, heme redox potentials, reaction kinetics with AsA and the MDA radical, specific inhibition of electron transfer by DEPC treatment, and AsA-dependent transmembrane ferric reductase activity), supporting the notion that the Cecytb-2 protein functions as an AsA-specific transmembrane ferric reductase in *C. elegans*.

### 3.8. Localization of the Cecytb-2 Protein

It is known that Dcytb proteins are expressed in the intestines of higher animals and function as ferric reductases to reduce ferric ions to the ferrous state. The ferrous ions thus formed are transported into the cytosol by DMT (divalent metal transporter), which is localized on the same luminal side of intestinal cells. It must be noted that the SMF-3 protein, a homolog of DMT-1 in *C. elegans* [51], was also found to be specifically expressed on the surface of the luminal side of intestinal cells [61,62]. To verify the localization of *Cecytb-2* gene expression, we performed a whole-mount in situ hybridization analysis using the *Cecytb-2* mRNA antisense DIG-labeled RNA probe. The fluorescent image corresponding to the Cecytb-2 mRNA was distributed in intestinal cells (Figure 7A). To further determine the cellular localization of the Cecytb-2 protein, we conducted whole-mount immunostaining using a purified polyclonal anti-Cecytb-2 antibody, specific to the C-terminal region of the Cecytb-2 protein (i.e., PVPWRREKTPDELK). The whole-mount immunostaining revealed that the Cecytb-2 protein was expressed exclusively in the intestines. Careful examination indicated that only the intestinal lumens were stained by the anti-Cecytb-2 antibody (Figure 7C). The specificity of the purified anti-Cecytb-2 antibody was examined by Western blotting analysis for the total proteins obtained from an adult *C. elegans*. Only one protein band with an apparent molecular mass of 28 kDa, very close to the theoretical molecular mass of the native Cecytb-2 protein (28.127 kDa), was detected (Figure 7B). This indicated that the purified antibody had a high specificity toward the Cecytb-2 protein. The specific localization of the Cecytb-2 protein in intestinal tissues was consistent with the notion that the Cecytb-2 protein functions as an AsA-specific transmembrane ferric reductase.

### 3.9. RNAi Experiments

To confirm our hypothesis, we performed RNAi experiments. If the Cecytb-2 protein indeed functions as a transmembrane ferric reductase in collaboration with the SMF-3 protein for the acquisition of iron into the body of *C. elegans*, knockdown of *Cecytb-2* gene expression (Appendix A) would cause a significant inhibition of the iron absorption ability and, accordingly, the growth of such iron-deficient worms would be depressed significantly [51]. However, for the control N2 worms grown under normal NGM plates, knockdown of the *Cecytb-2* gene by the feeding RNAi method did not show any appreciable changes in their phenotypes, such as their life span (Appendix A), motility (Appendix A), memory (Appendix A), growth (Figure 8A, left panels), and the number of hatched eggs (Figure 8C). We thought that the knockdown worms could somehow maintain their normal growth by utilizing small but sufficient amounts of ferrous ions in their environments (i.e., in the NGM plates). To prove this possibility, we removed the environmental ferrous ions by using a specific ferrous ion chelator, BP (2,2′-dipyridyl) [51,63], in the NGM plates with final concentrations of 20–60 μM. The addition of BP (60 μM) alone to the NGM plate significantly affected the number of hatched eggs from the control N2 worms (Figure 8B), but it did not affect the growth of the control N2 worms at all (Appendix A). However, the Cecytb-2-RNAi-treated N2 worms showed significant retardation in their growth (but not in their life span (Appendix A) when they were laid on the NGM plate containing BP (40 μM) (Figure 8A, left panels). Further, hatching of the eggs from the Cecytb-2-RNAi-treated N2 worms did not occur by the addition of BP (60 μM) (Figure 8C, right). These results indicated that the Cecytb-2 knockdown worms could not utilize environmental ferric ions sufficiently because of the suppression of ferric reductase activities when they were laid on NGM plates containing BP.

## 4. Discussion

AsA is required as a co-factor for various important enzymes for collagen and neurotransmitter synthesis (such as homologs of prolyl 4-hydroxylase, lysyl hydroxylase, dopamine β-hydroxylase, peptidylglycine α-amidating monooxygenase) in *C. elegans* [29] and, therefore, its presence in the *C. elegans* body has been assumed without any concrete evidence. However, recent studies have proven that *C. elegans* can biosynthesize AsA [29] with a biosynthetic pathway similar to that found in mammals [30]. The concentration level of AsA varies depending on the life stage of the organism [29]. Thus, based on our present results, we can conclude that the Cecytb-2 protein functions as an AsA-specific transmembrane ferric reductase on the luminal (or apical) side of the intestines in adult *C. elegans.* We reached this conclusion from two distinct approaches, i.e., at the molecular mechanistic level and at the physiological level.

In molecular mechanistic studies, we revealed that the recombinant Cecytb-2 protein exhibited characteristic UV–visible absorption spectra, consistent with its membership in the cytochrome *b*_561_ family (Figure 1B). Heme content analysis indicated two molecules of heme *b* per one protein molecule (Appendix A). The rapid electron-accepting ability from AsA to the oxidized heme centers (Figure 4, Appendix A) and its specific inhibition upon DEPC treatment without the loss of the heme *b* moiety (Figure 3D) were well conserved for the Cecytb-2 protein, as previously shown for other members of the cytochrome *b*_561_ family [59]. The EPR signals of the Cecytb-2 protein (g_z_ = 3.27 and g_z_ = 3.65) (Figure 2) were more similar to those of Dcytb than CGcytb, suggesting that Cecytb-2 and Dcytb have similar environments in their respective heme prosthetic group. Redox potential measurements of Cecytb-2 showed the closeness of its two midpoint potentials to each other (+120 mV for heme *b*_H_ and +92 mV for heme *b*_L_) (Figure 3B), again very similar to those of human Dcytb (two potentials with +80 mV ± 30 mV) [23]. These results suggest the possibility that members of the Dcytb sub-group have very close midpoint potentials for their two heme centers in common, and this nature may be related to their efficiency in ferric reductase activity and the lowered reactivity toward the MDA radical (Figure 5).

Interestingly, DEPC treatment of Cecytb-2 caused a significant lowering of the midpoint potential for the heme *b*_L_ center. Although we considered that the major causes of the inhibition of electron transfer from AsA to the heme *b*_L_ center upon DEPC treatment were due to the *N*-carbethoxylations of His101 and Lys98, as proved by our present MS/MS analysis, the exact mechanical basis of the inhibition is still not clear, other than the crowding effect caused by *N*-carbethoxyl groups. However, the lowered midpoint potential of the heme *b*_L_ center upon the *N*-carbethoxylation of the heme-coordinating imidazole group (of His101) could have some roles in the inhibition of the electron transfer from AsA. In addition, removal of a positive charge upon the *N*-carbethoxylation of Lys98 residue, which is located very near the heme *b*_L_ center, might also render the lowering of its midpoint potential to heme *b*_L_, as well as inhibition of the electron transfer from AsA.

It must be noted at this point that the positions of the heme *b*_L_ and *b*_H_ centers in the cytochrome *b*_561_ protein family have been very controversial, with our original model assigning a cytosolic His pair (His101/His174 in the Cecytb-2 case) to be coordinated to the heme *b*_L_ center [33,58,59,64], versus a later model assigning an extracellular His pair (His67/His135 in the Cecytb-2 case) to be coordinated to the heme *b*_L_ center [54,57,65,66,67]. The key issue was whether it is heme *b*_H_ or heme *b*_L_ that is exposed to the cytoplasm and has access to cytosolic AsA. We believe that our original proposal, “heme *b*_L_ on the cytoplasm side”, was established in the present study, based on two experimental observations (the MS/MS analysis and the redox potential measurements of the DEPC-treated Cecytb-2 protein). Supporting our conclusion, we observed a drastic lowering of the midpoint potential of the heme *b*_L_ center and a significant decrease in reactivity with AsA for three K98 mutants of Cecytb-2 (Unpublished results, available upon request from the corresponding author). This conclusion is consistent with the view that physiological electron transfer occurs from a low-potential center to a high-potential center. AsA would donate its electron to the low-potential heme *b*_L_ center on the cytosolic side, and then the electron would move to the high-potential heme *b*_H_ center on the extracellular (or apical) side in the cytochrome *b*_561_ protein family, and, therefore, this process would not require the presence of membrane potential [57].

Our proteoliposome experiment (Figure 6) indicated that the reconstituted Cecytb-2 protein could donate electrons to ferric ions via a transmembrane electron transfer reaction, proving that the Cecytb-2 protein is an AsA-specific transmembrane ferric reductase and, therefore, it might have an interacting site with a ferric ion(s) on the extracellular (or luminal or apical) side. The distribution of negatively and positively charged amino acid residues in the extracellular loops may be crucial for directing ferric ions to the active site, facilitating an efficient reduction process. A recent X-ray crystal study on human Dcytb suggested that several negatively charged residues (E36, D41, E106, D189, and E197) (Appendix A) on the extracellular (luminal or apical) surface might have such roles. A closer look at these three extracellular loops of the Cecytb-2 protein in comparison with those of human Dcytb and those from various higher animal species indicated that the Cecytb-2 protein does have a distinctly different distribution of these charges (Appendix A). On the other hand, excess positive charges would facilitate the access of a negatively charged MDA radical to the catalytic site. Consistent with this view, the h101F6 protein, which showed the highest activity with the MDA radical so far measured [18], has a very scarce distribution of negative charges on the intravesicular (or extracellular) side (Appendix A). Thus, the presence of both ferric reductase activity and MDA radical reductase activity in the Cecytb-2 protein, albeit with a lower efficiency for the latter, is due to its intermediate nature within the cytochrome *b*_561_ protein family. It must be noted, however, that the distribution of the charges alone cannot explain the presence and strength of the activities, as in the case of CGcytb, where many negative charges are present on the extracellular (intravesicular) side (Appendix A) but exhibiting a rather high MDA radical reductase activity [45]. To clarify the nature of these electron transfer activities on the extracellular side, detailed calculations based on X-ray structural data [24,60] may be required.

Based on the in situ hybridization and immunohistochemical staining experiments, we found that the Cecytb-2 protein was expressed exclusively on the luminal side of the intestines (Figure 7). The SMF-3 protein, a homolog of DMT1 in *C. elegans*, is known to function as a divalent metal transporter and is expressed on the luminal side of intestines [51,61]. Co-localizations of the SMF-3 [61] and Cecytb-2 proteins (present study) indicated a close functional relationship between these two. In mammals, Dcytb and DMT1 are expressed at the apical (luminal) surface of digestive organs [22], where the DMT1 protein functions as the divalent metal transporter to uptake iron but only in the ferrous state [21]. The ferrous ions are provided by the Dcytb protein by the reduction of environmental ferric ions using cytosolic AsA as the source of electrons [22]. Based on these pieces of evidence, we can conclude that Cecytb-2 protein functions as a bona fide transmembrane ferric reductase to provide ferrous ions to the SMF-3 protein on the same luminal side of the intestines for the acquisition of iron.

The Cecytb-2 knockdown worms by RNAi did not exhibit any unusual phenotypes when grown under normal conditions. However, when the environmental ferrous ion was limited by a rapidly penetrating lipophilic Fe^2+^ chelator, BP (i.e., NGM plus 60 μM BP), the number of hatched eggs decreased significantly by RNAi (Figure 8C). The Cecytb-2 knockdown worms also showed significant impairment in their growth on the NGM plate containing BP (40 μM) (Figure 8A,C). On the other hand, when we examined the effect of RNAi under low ferrous ion conditions (i.e., NGM plate without BP), we did not observe any significant defect in their growth (Appendix A). How could the Cecytb-2 knockdown worms acquire enough amounts of iron under the low-ferrous condition without sufficient ferric reductase activities? There are two major iron acquisition pathways in higher animals: the direct acquisition of iron in a ferrous state by a divalent transporter residing on the intestinal enterocyte cell membranes, followed by circulation in the body by transferrin/transferrin receptor system, or the uptake of heme by HCP1 on the same enterocyte cell membranes [68], followed by intracellular enzymatic degradation of heme (mostly by heme oxygenase; HO) to produce ferrous iron. It must be noted that *C. elegans* is a heme auxotroph and, therefore, has specific trafficking machinery to utilize environmental heme [69]. Accordingly, even without ferric reductase activity, *C. elegans* can survive by direct utilization of the acquired heme. Indeed, it was reported that heme-regulated proteins, HRGs, might function as heme transporters at the lumen of the digestive organs [70,71,72]. However, the genome of *C. elegans* does not contain any orthologs corresponding to HO [73], suggesting that heme degradation in *C. elegans* is catalyzed by a HO with very low sequence homology or by an entirely novel enzyme [73], such as glutathione reductase [74]. Thus, there is a possibility that in *C. elegans*, the system for heme utilization to support the demand for non-heme iron is very vulnerable or its efficiency is very low. In such cases, strict limitation of ferrous ions from the environment would cause severe impairment in developmental growth (particularly during hatching). In accordance with this proposal, it was reported previously that knockout worms for hypoxia-inducible factor 1 (HIF-1) cultured under the ferrous deficient conditions showed a very similar phenotype to that observed in the present study, i.e., significant developmental delay [51], suggesting that the pool of chelatable ferrous ion that constitutes less than 5% of total cellular iron is very important for the usual developmental functions in worms [51]. Therefore, the ferric reductase activity of Cecytb-2 may have prime importance for the early stage of the worm’s life, where the heme acquisition system and its degradation system do not function or are inadequate.

It is known that the HIF-1 protein promotes the expression of iron transporter SMF-3 to facilitate iron acquisition through the presence of the IDE (iron-dependent enhancer) sequence in the promoter region, where several HRE (hypoxia-response element) as the binding site for the HIF-1 protein and GATA sequences are distributed [51]. An intestine-specific transcription regulator protein, ELT-2, is known to bind the GATA sequence [75,76]. Similarly, human HIF-1 is known to regulate the expression of DMT and Dcytb [77,78] through the presence of IDE sequences in their promoter regions, where several HREs are distributed as the binding sites for the HIF-1 protein [79,80]. Therefore, deletion of the *hif-1* gene would cause insufficient expression of iron transporters and excessive expression of the ferritin protein. Then, these events would induce a decrease in the level of iron utilization. The occurrence of very similar phenotypes for both the Cecytb-2 knockdown worms and HIF-1 knockout worms suggested that the coupling of the Cecytb-2 protein to the iron uptake system may also be regulated at the gene expression level. To support this possibility, we searched a promoter region (up to 3000 bp in 5′ upstream) of the *Cecytb-2* (*F39G3.5*) gene. Interestingly, we found at least four HRE sequences and nine GATA sequences in this region (Appendix A).

Taken together, our present results indicate that the Cecytb-2 protein is a genuine homolog of the human Dcytb protein and functions as an AsA-specific transmembrane ferric reductase on the luminal side of the intestines. Therefore, the molecular mechanism of iron acquisition is well conserved between mammals and nematodes. Furthermore, the ferric reductase activity of the Cecytb-2 protein plays a crucial role in the early development (particularly during hatching) of *C. elegans*. This is the first report on the homologs of cytochrome *b*_561_ in *C. elegans*, examining both biochemical and physiological levels. The present study proved that *C. elegans* is a suitable model for elucidating the physiological functions of various membrane proteins, including cytochrome *b*_561,_ present in higher organisms.

## 5. Conclusions

Our experimental results on EPR, heme redox potentials, and reaction kinetics with AsA indicated that the Cecytb-2 protein was more similar to human Dcytb. Reconstituted Cecytb-2 proteins in AsA-entrapped sealed vesicle membranes showed significant transmembrane ferric reductase activity. In situ hybridization and immunohistochemical analyses revealed that Cecytb-2 is expressed in the intestinal lumens of *C. elegans*. Knockdown of the Cecytb-2 gene expression in N2 worms resulted in significant suppression of their growth under ferrous ion-deficient conditions. Thus, the ferric reductase activity of Cecytb-2 appears to play a role in iron acquisition and is crucial for normal growth under low-ferrous conditions, confirming that Cecytb-2 is a true Dcytb homolog in *C. elegans*.

## Figures and Tables

**Figure 1 biomolecules-15-01385-f001:**
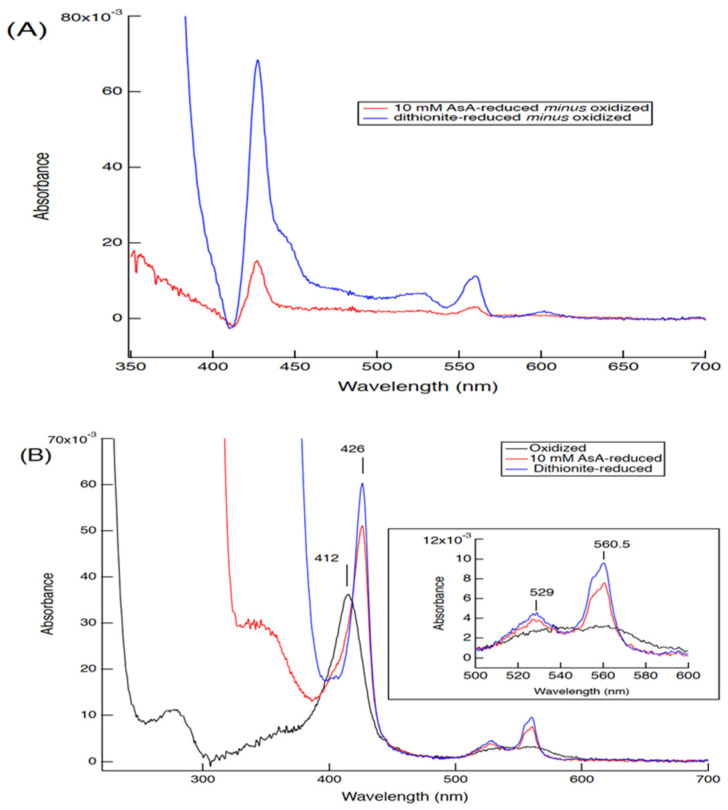
UV–visible absorption spectra of the heterologously expressed Cecytb-2-H_6_ protein. (**A**) The redox difference visible absorption spectra of the microsomal fraction (10 mM AsA (red line) or dithionite (blue line) reduced form *minus* air-oxidized form, containing 380 nM Cecytb-2 protein). (**B**) The purified Cecytb-2-H_6_ protein in air-oxidized form (240 nM) in 50 mM potassium phosphate buffer, pH 7.4, containing 10% (*v*/*v*) glycerol and 1% (*w*/*v*) OG, is indicated as a black line. The reduced forms with 10 mM AsA and with dithionite are indicated in red and in blue, respectively. The region from 500 nm to 600 nm is enlarged as the inset.

**Figure 2 biomolecules-15-01385-f002:**
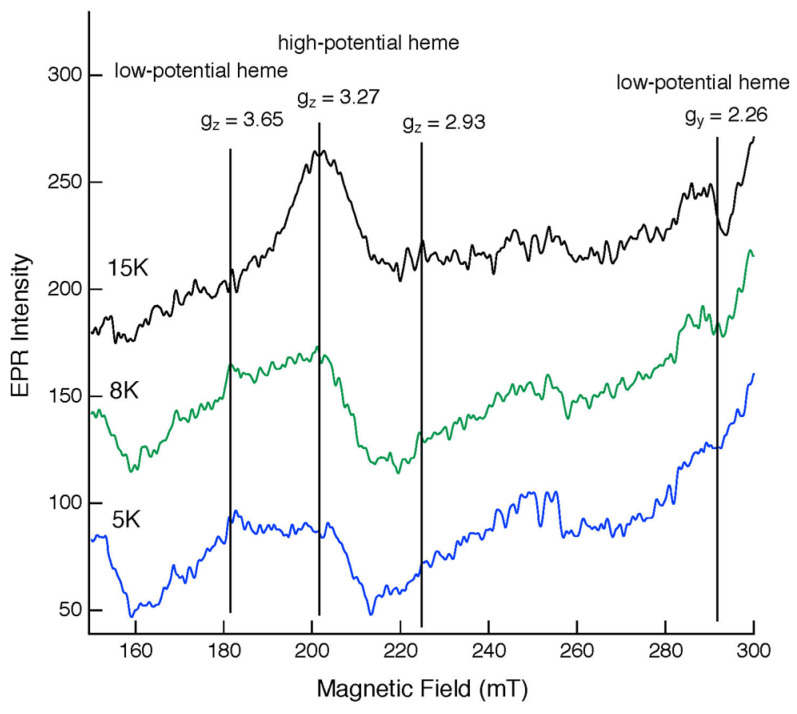
X-band EPR spectra of the purified Cecytb-2-H_6_ protein in an air-oxidized state. The purified Cecytb-2-H_6_ protein in an air-oxidized state (60 μM in 50 mM potassium phosphate buffer, pH 7.4, containing 10% (*v*/*v*) glycerol and 1% (*w*/*v*) OG) was measured in a frozen state at different temperatures: 5 K (blue), 8 K (green), and 15 K (black). The calculated g-values are indicated along each vertical line. The assignment of the EPR signals to two heme species (low-potential heme *b*_L_ and high-potential heme *b*_H_) is based on [59].

**Figure 3 biomolecules-15-01385-f003:**
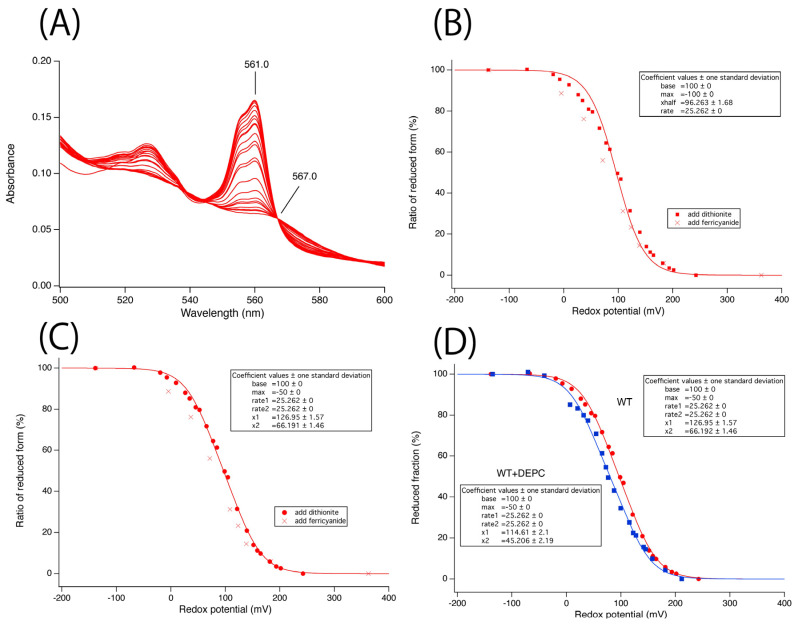
Redox titration of the purified Cecytb-2-H_6_ protein (final, ≈5 μM) in 50 mM potassium phosphate buffer (pH 7.4), 0.1% (*w*/*v*) DDM, and 10% (*v*/*v*) glycerol at 20 °C. (**A**) Absolute spectral changes in the visible region during the reductive titration with sodium dithionite. (**B**) Fitting of the data points by a Nernst equation with a single redox center. (**C**) Fitting of the data points by a Nernst equation with double redox centers. Only the data points for the reductive titration phase (red solid circles) were analyzed. Data points for the oxidative titration phase (solid crosses) are included for comparative purposes. (**D**) Effects of the DEPC treatment of Cecytb-2-H_6_ on its midpoint potentials. Purified Cecytb-2-H_6_ protein in an oxidized state was treated with 5 mM DEPC for 30 min; then, its heme redox potentials were measured in a similar manner. The fitting analyses of the WT + DEPC sample (solid blue squares) indicated that the heme *b* center with a lower midpoint potential (heme *b*_L_) showed a much larger shift (21 mV shift from +67 to +46 mV) than the higher redox heme *b* center (heme *b*_H_) (12 mV shift from +126 to +114 mV).

**Figure 4 biomolecules-15-01385-f004:**
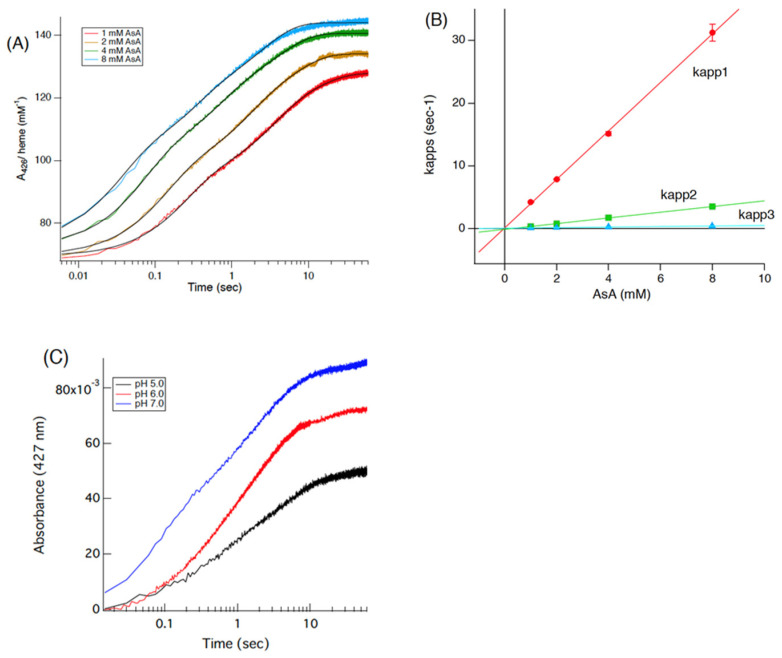
Stopped-flow kinetic analyses on the electron transfer reaction of the purified Cecytb-2-H_6_ protein with AsA. (**A**) Oxidized Cecytb-2-H_6_ protein (final 1 μM in 50 mM potassium phosphate buffer, pH 7.0, containing 10% (*v*/*v*) glycerol and 1% (*w*/*v*) OG) was mixed with an equal volume of different concentrations of AsA (final 1, 2, 4, 8 mM), and the following absorption changes at 426 nm were measured. The data were fitted using a linear combination of three exponential functions for both panels (**A**,**B**), as described in the experimental procedures. (**B**) AsA concentration-dependent changes of three apparent rate constants obtained from the analyses of the data shown in panel (**B**) using a linear combination of three exponential functions. (**C**) The reduction of oxidized Cecytb-2-H_6_ protein (final, 1 μM in 50 mM potassium phosphate buffer for pH 6.0 and 7.0 and in 50 mM sodium acetate buffer for pH 5.0, containing 10% (*v*/*v*) glycerol and 1% (*w*/*v*) OG) with AsA (final 2 mM) was conducted by a stopped-flow method (in a 1:1 volume ratio) and was monitored by the absorption change at 427 nm at three different pH values (pH 5.0 (black), 6.0 (red), and 7.0 (blue)).

**Figure 5 biomolecules-15-01385-f005:**
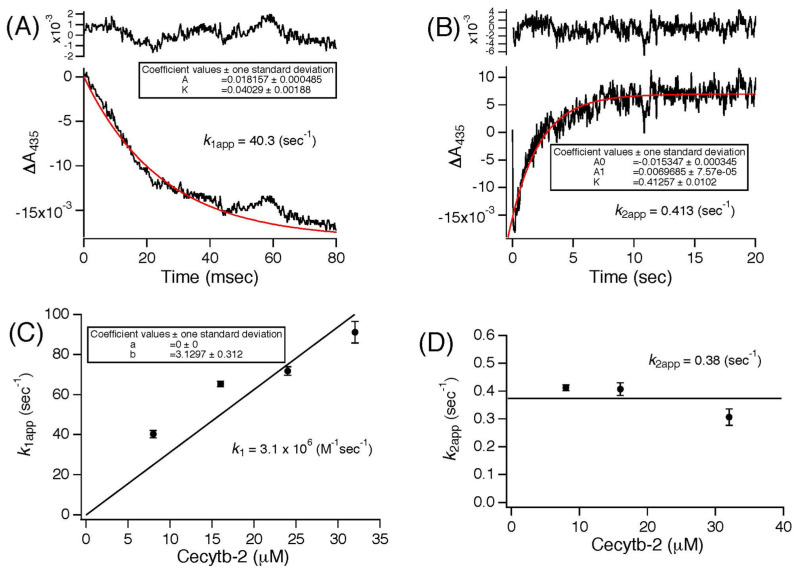
Electron transfer reactions of the Cecytb-2-H_6_ protein with the MDA radical and AsA were analyzed by a pulse radiolysis method. (**A**) The purified Cecytb-2-H_6_ protein (final, 8 μM) and AsA (final, 10 mM) were dissolved in 10 mM potassium phosphate buffer (pH 7.0) containing 1% (*w*/*v*) OG, 10% (*v*/*v*) glycerol, and 300 mM NaCl. The rapid oxidation reaction of Cecytb-2-H_6_ with the pulse-generated MDA radical (1.5 μM) was measured by the absorbance change at 435 nm in the time domain of 80 ms. (**B**) The following slower re-reduction of Cecytb-2-H_6_ by AsA was measured in the time domain of 20 s. (**C**) The apparent rate constants for the fast oxidation phase (*k*_1app_) were measured at four different concentrations of Cecytb-2-H_6_ samples (8, 16, 24, 32 μM) and were plotted against the protein concentration. The second-order rate constant was calculated from the slope as 3.1 × 10^6^ M^−1^s^−1^. (**D**) The apparent rate constants for the slow re-reduction phase (*k*_2app_) are plotted against the Cecytb-2-H_6_ concentration, showing no appreciable dependency.

**Figure 6 biomolecules-15-01385-f006:**
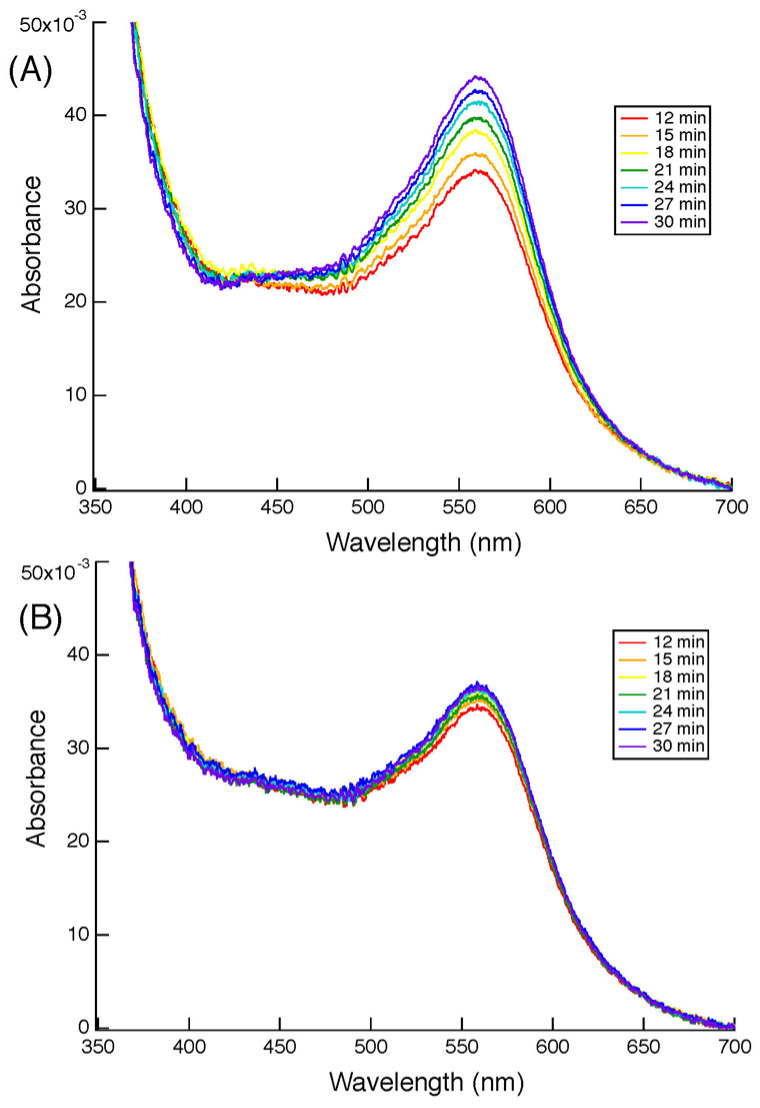
PDTS assay using Cecytb-2-H_6_ reconstituted proteoliposomes with entrapped AsA (100 mM) to determine the transmembrane ferric reductase activity of Cecytb-2-H_6_. (**A**) The build-up of absorption at 562 nm indicates the formation of ferrous–PDTS complexes, which were produced by the transmembrane ferric reductase activity of Cecytb-2-H_6_ reconstituted in the liposomal membranes. The externally added ferric substrate, FeCl_3_ (200 μM), was reduced by Cecytb-2-H_6_ using electrons from AsA (100 mM), which was entrapped in the liposomes. The resulting ferrous ion was chelated with PDTS (100 μM). (**B**) A control assay using liposomes prepared without the addition of any proteins but entrapped with AsA (100 mM) was conducted by mixing with ferric ions and PDTS.

**Figure 7 biomolecules-15-01385-f007:**
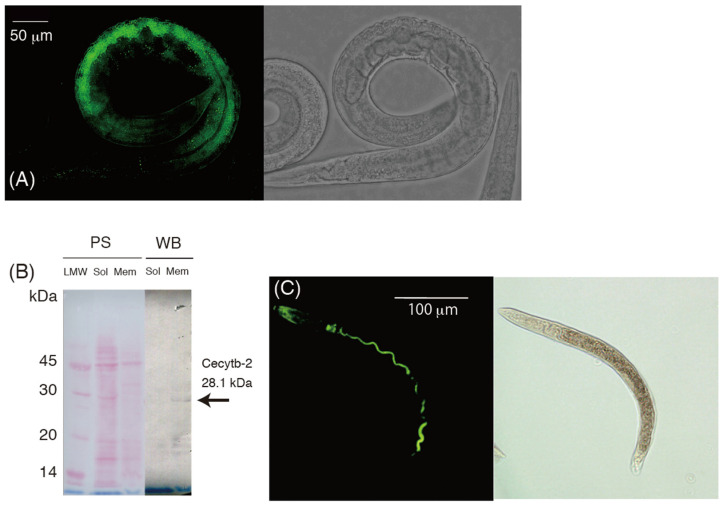
Localized expression of Cecytb-2 mRNA and the Cecytb-2 protein in adult *C. elegans*. (**A**) In situ hybridization showed that only intestinal cells were stained with the Cecytb-2 antisense probes. After the hybridization with a DIG-labeled Cecytb-2 antisense probe, the hybridized probes were detected by anti-DIG rabbit IgG as the primary antibody and Alexa fluor 488-conjugated anti-rabbit IgG goat IgG as the secondary antibody. The fluorescence was excited at 488 nm and detected at 530 nm. In the right panel, a DIC image is shown. (**B**) Western blotting analysis of Cecytb-2 protein expression in adult *C. elegans*. The soluble fraction (Sol) (15 μL) and membrane protein fraction (Mem) (5 μL) extracted from adult *C. elegans* and a molecular weight marker (LMW) were each applied onto an SDS-PAGE gel, electrophoresed, and blotted onto a nitrocellulose membrane. The left panel shows the protein bands stained with Ponceau S (PS). The detection of the Cecytb-2 protein was performed using a diluted anti-Cecytb-2 rabbit antibody as the primary antibody and a diluted HRP-conjugated anti-rabbit IgG goat antibody as the secondary antibody. The Cecytb-2 protein was detected by the addition of 4-chloro-1-naphthol and hydrogen peroxide, as shown in the right panel (WB). The arrow indicates the endogenous form of the Cecytb-2 protein (28.1 kDa) in *C. elegans*. (**C**) Immunostaining of the Cecytb-2 protein in adult *C. elegans* using anti-Cecytb-2 rabbit antibody and Alexa fluor 488-conjugated anti-rabbit IgG goat antibody, indicating that the Cecytb-2 protein is localized at intestinal lumens. The fluorescence was excited at 488 nm and detected at 530 nm. In the right panel, a usual light microscope image is shown.

**Figure 8 biomolecules-15-01385-f008:**
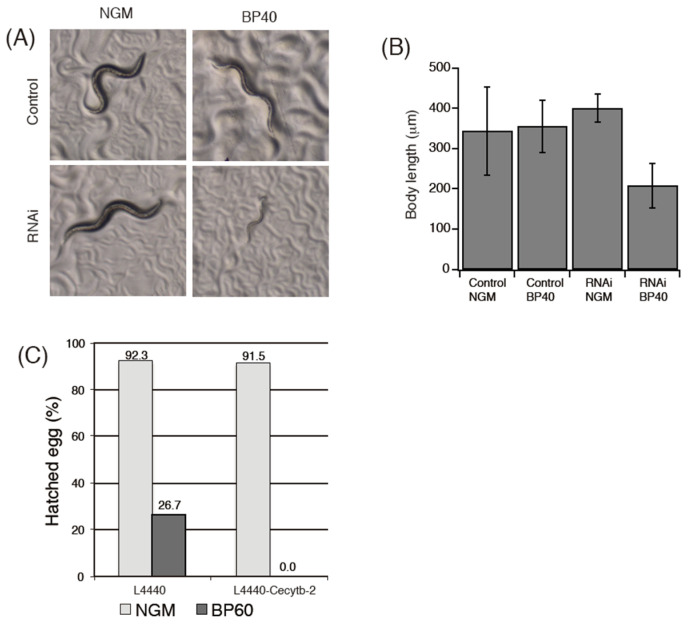
Effects of Cecytb-2 RNAi treatment on the phenotypes of *C. elegans* and on the hatching of their eggs. (**A**) Control worms (fed with *E. coli* transformed with an L4440 empty vector) and RNAi-treated worms (fed with *E. coli* transformed with an L4440/Cecytb-2 vector) were each grown on NGM plates with or without 40 μM BP (BP40) for 55 h at 30 °C. The RNAi–treated worms grown on the NGM plates did not show any impairments in their growth (left panels), but the RNAi–treated worms grown on the NGM plates with BP40 showed impaired growth (lower right panel). (**B**) Statistical evaluation of body length for the RNAi-treated worms (or control worms) grown on the NGM plates with or without BP40 for 55 h at 30 °C (N = 31). (**C**) Effects of the RNAi treatment on the number of hatched eggs. The synchronized L4 worms were incubated for 24 h on the NGM plates supplemented with or without 60 μM BP. The number of laid eggs was counted, and they were incubated for an additional 24 h. Then, the hatched larvae were removed, and the number of remaining eggs was counted.

## Data Availability

The original contributions presented in this study are included in the article/Appendix A. Further inquiries can be directed to the corresponding author.

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
