# Peer review of "Cecytb-2, a Cytochrome b561 Homolog, Functions as an Ascorbate-Specific Transmembrane Ferric Reductase at Intestinal Lumens of Caenorhabditis elegans"

_biomolecules, 2025, doi:10.3390/biom15101385_

Round 1
Reviewer 1 Report
Comments and Suggestions for Authors
In this manuscript, the authors presented the evidence suggesting that the cytochrome b561 homolog (Cecytb-2) from E. elegans is an ascorbate-specific transmembrane ferric reductase at the intestinal lumen. The results were quite comprehensive with biochemical characterizations of purified Cecytb-2. immunohistochemical analysis, and the knockdown phenotypes. The manuscript was well written. However, there were a few points that the authors might consider for the revision:
- In the abstract, the authors stated that “Cecytb-2 accepts electrons from AsA (ascorbate) very rapidly but donates them to monodehydroacobate radical (MDA) with a much lower reactivity than other cytochrome b561”. This statement is a little confusion as Figure 5 seemed to clearly indicate that reduction of Cecytb-2 by AsA is much slower than oxidation of Cecytb-2 by MDA. Compared with other cytochrome b561, Cecytb-2 seems to be slower to reduce MDA. Presumably, in the presence of ferric iron, the reduced Cecytb-2 will also be oxidized. Will ferric iron effectively compete with MDA for oxidation of the reduced Cecyb-2?
- Figure 6 indicates that the liposome-entrapped ASA reduced a significant amount of ferric iron without Cecytb-2. The authors stated that “ferric iron was reduced by a very slow leakage of from entrapped AsA” without showing the kinetics of AsA leakage. Would it be possible to show the oxidation and reduction kinetics of the liposome-bound Cecytb-2 in the presence of ferric iron and AsA to show the ferric reductase activity of Cecytb-2 clearly?
- Line 650, missing part of a sentence.
- Discussion section could be shortened (without repeating the result section).
Author Response
In this manuscript, the authors presented the evidence suggesting that the cytochrome b561 homolog (Cecytb-2) from E. elegans is an ascorbate-specific transmembrane ferric reductase at the intestinal lumen. The results were quite comprehensive with biochemical characterizations of purified Cecytb-2. immunohistochemical analysis, and the knockdown phenotypes. The manuscript was well written.
(Answer and response) Thank you very much for very kind comments.
However, there were a few points that the authors might consider for the revision:
- In the abstract, the authors stated that “Cecytb-2 accepts electrons from AsA (ascorbate) very rapidly but donates them to monodehydroacobate radical (MDA) with a much lower reactivity than other cytochrome b561”. This statement is a little confusion as Figure 5 seemed to clearly indicate that reduction of Cecytb-2 by AsA is much slower than oxidation of Cecytb-2 by MDA. Compared with other cytochrome b561, Cecytb-2 seems to be slower to reduce MDA. Presumably, in the presence of ferric iron, the reduced Cecytb-2 will also be oxidized. Will ferric iron effectively compete with MDA for oxidation of the reduced Cecyb-2?
(Answer and response) We are very sorry to make confusion. The statement “Cecytb-2 accepts electrons from AsA (ascorbate) very rapidly but donates them to monodehydroacobate radical (MDA) with a much lower reactivity than other cytochrome b561” is meaning as “Cecytb-2 accepts electrons from AsA (ascorbate) very rapidly as found for other cytochrome b561 but donates them to monodehydroacobate radical (MDA) with a much lower reactivity than other cytochrome b561”. However, to remove the confusion, we changed the abstract significantly, as shown below (highlighted in yellow).
Fast kinetic experiments using pulse-radiolysis and stopped-flow techniques showed that Cecytb-2 donates electrons to monodehydroascorbate radical with a much lower reactivity than other cytochromes b561 but can accepts electrons from ascorbate (AsA) rapidly as for other members.
About the reactivity to MDA. In the pules radiolysis experiment, ferric ion was not included in the buffer system (as indicated in Figure 5, and in our previous experiments for other cytochrome b561) and, therefore, the reactivity of reduced heme b with MDA would be its intrinsic one.
- Figure 6 indicates that the liposome-entrapped ASA reduced a significant amount of ferric iron without Cecytb-2. The authors stated that “ferric iron was reduced by a very slow leakage of from entrapped AsA” without showing the kinetics of AsA leakage. Would it be possible to show the oxidation and reduction kinetics of the liposome-bound Cecytb-2 in the presence of ferric iron and AsA to show the ferric reductase activity of Cecytb-2 clearly?
- (Answer and response) The amount of liposome-bound Cecytb-2 was very low, far below the level of our detection by usual spectrometric methods. Therefore, we did not measure the oxidation and reduction kinetics of the liposome-bound Cecytb-2 in the presence of ferric iron and AsA,
- Line 650, missing part of a sentence.
(Answer and response) The incomplete sentence (at Line 650) was removed in the revised manuscript.
- Discussion section could be shortened (without repeating the result section).
- (Answer and response) In the revised manuscript, discussion section was shortened following the suggestions of reviewer 1. We have tried to remove the repeating of the result section.
Reviewer 2 Report
Comments and Suggestions for Authors
Review on the manuscript of Miura M et al., (biomolecules-3863761): “Cecytb-2, a cytochrome b561 homolog, functions as an ascorbate-specific transmembrane ferric reductase at intestinal lumens of Caenorhabditis elegans”.
In this study, the Authors provide a detailed characterization of the biochemical and biophysical characteristics of the C. elegans Cecytb-2 protein, as well as its physiological role as a transmembrane ferric reductase specific for ascorbate. The Authors show that purified recombinant Cecytb-2 was rapidly reduced by ascorbate and displays typical cytochrome b561 spectral properties, with redox features similar to Dcytb. Kinetic studies showed that Cecytb-2 efficiently accepts electrons from ascorbate but transfers them to the monodehydroascorbate radical more slowly than other cytochromes b561. Furthermore, DEPC treatment inhibited electron acceptance and lowered the heme bL midpoint potential, which was hypothesized to be primarily due to N-carbethoxylation of Lys98 and His101. Moreover, reconstituted Cecytb-2 in vesicles exhibited significant transmembrane ferric reductase activity. Additionally, Cecytb-2 mRNA and protein were localized in intestinal cells, and Cecytb-2 knockdown impaired worm growth under low ferrous conditions. Based on these results, the Authors suggest Cecytb-2 as a true Dcytb ortholog, contributing to iron acquisition from intestinal lumen and is essential for the normal growth of C. elegans.
Overall, I find the study of the Cecytb-2 protein in C. elegans highly interesting, as it clarifies whether this protein is a true ortholog of Dcytb, a key ferric reductase in human iron absorption. Moreover, by characterizing its biochemical properties and physiological role in C. elegans, the study may provide insights into conserved mechanisms of intestinal iron uptake. This knowledge not only supports the utility of C. elegans as a model to study iron deficiency and related disorders but may also help define potential strategies to overcome these conditions.
I believe the Authors have addressed the primary question proposed. Below, I indicate the issues identified in the current version of the manuscript. I hope the Authors find the following comments and suggestions helpful.
1 - I kindly suggest that the Authors consider reorganizing the abstract. In its current form, it does not provide background information to introduce the topic or highlight its significance.
2 - In line 197, I kindly suggest that the Authors clarify whether “EDTA” was intended to be “DEPC”.
3 - I kindly suggest that the Authors consider increasing the overall size of the figures to make the information easier to visualize.
4 - In lines 597-599, the Authors mention that “The apparent rate constants of the re-reduction reaction (k2app) were found to be almost independent of the protein concentration with a mean value of 0.36 (s-1) (Fig. 5D)”. However, in Figure 5D, the value indicated corresponds to 0.38 (s-1). I kindly recommend that the Authors clarify this point.
5 - In line 645, where it reads “Fig. 12(A)” should be “Fig. S12(A)”.
6 - I kindly suggest that the Authors include a description of the data shown in Figure 8C at the end of the Results section, as it is not currently mentioned in the manuscript.
7 - The Authors suggest that the inhibition of electron transfer from AsA to the heme bL center upon DEPC treatment is primarily due to N-carbethoxylation of His101 and Lys98, as indicated by the MS/MS analysis. While this is a reasonable hypothesis, it remains suggestive. A practical approach to test it would be to generate a recombinant Cecytb-2 protein with these two residues mutated and evaluate it using the same assay. This would provide a direct and straightforward way to confirm the proposed mechanism.
8 - In line 878, please replace “purported” with “supported.
Author Response
In this study, the Authors provide a detailed characterization of the biochemical and biophysical characteristics of the C. elegans Cecytb-2 protein, as well as its physiological role as a transmembrane ferric reductase specific for ascorbate. The Authors show that purified recombinant Cecytb-2 was rapidly reduced by ascorbate and displays typical cytochrome b561 spectral properties, with redox features similar to Dcytb. Kinetic studies showed that Cecytb-2 efficiently accepts electrons from ascorbate but transfers them to the monodehydroascorbate radical more slowly than other cytochromes b561. Furthermore, DEPC treatment inhibited electron acceptance and lowered the heme bL midpoint potential, which was hypothesized to be primarily due to N-carbethoxylation of Lys98 and His101. Moreover, reconstituted Cecytb-2 in vesicles exhibited significant transmembrane ferric reductase activity. Additionally, Cecytb-2 mRNA and protein were localized in intestinal cells, and Cecytb-2 knockdown impaired worm growth under low ferrous conditions. Based on these results, the Authors suggest Cecytb-2 as a true Dcytb ortholog, contributing to iron acquisition from intestinal lumen and is essential for the normal growth of C. elegans.
Overall, I find the study of the Cecytb-2 protein in C. elegans highly interesting, as it clarifies whether this protein is a true ortholog of Dcytb, a key ferric reductase in human iron absorption. Moreover, by characterizing its biochemical properties and physiological role in C. elegans, the study may provide insights into conserved mechanisms of intestinal iron uptake. This knowledge not only supports the utility of C. elegans as a model to study iron deficiency and related disorders but may also help define potential strategies to overcome these conditions. I believe the Authors have addressed the primary question proposed.
(Answer and response) Thank you very much for very kind comments.
Below, I indicate the issues identified in the current version of the manuscript. I hope the Authors find the following comments and suggestions helpful.
1 - I kindly suggest that the Authors consider reorganizing the abstract. In its current form, it does not provide background information to introduce the topic or highlight its significance.
(Answer and response) We made reorganization of the abstract to provide background information and for enhancing the significance of our results. Resultant abstract is as follows (modified part is highlighted in yellow).
One of cytochrome b561 family member in C. elegans, named as Cecytb-2, was investigated. Purified recombinant Cecytb-2 showed typical visible absorption spectra, EPR signals, and redox midpoint potentials, very similar to those of human Dcytb, which is responsible for intestinal iron acquisition by its ferric reductase activity. Fast kinetic experiments using pulse-radiolysis and stopped-flow techniques showed that Cecytb-2 donates electrons to monodehydroascorbate radical with a much lower reactivity than other cytochromes b561 but can accepts electrons from ascorbate (AsA) rapidly as for other members. DEPC-treatment of Cecytb-2 caused significant inhibition of the electron acceptance from AsA and a lowering of midpoint potential of heme bL. MS/MS MASCOT analyses verified that N-carbethoxylations of conserved Lys98 and heme bL axial His101 residues on the cytosolic side were major causes of the inhibition. Reconstituted Cecytb-2 in sealed vesicle membranes, in which AsA was entrapped, showed a significant transmembrane ferric reductase activity. In situ hybridization analysis revealed that Cecytb-2 mRNA distributed in intestinal cells. Immunohistochemical analysis indicated that Cecytb-2 resided at intestinal lumens. Knock-down of the Cecytb-2 gene expression in N2 worms indicated a significant suppression in growth at ferrous ion deficient conditions. Thus, the ferric reductase activity conferred by Cecytb-2 seemed to participate for their iron acquisition and is very important for their normal growth at low ferrous conditions, confirming that Cecytb-2 is a genuine Dcytb homolog in C. elegans.
2 - In line 197, I kindly suggest that the Authors clarify whether “EDTA” was intended to be “DEPC”.
(Answer and response) This was our mistake. EDTA was actually DEPC, In the revised manuscript, this part was changed as follows. “(final 0.5 mM by addition of 30 mM of DEPC in ethanol)”
3 - I kindly suggest that the Authors consider increasing the overall size of the figures to make the information easier to visualize.
(Answer and response) In the revised manuscript, we increased the size of the figures to make the information easier to visualize.
4 - In lines 597-599, the Authors mention that “The apparent rate constants of the re-reduction reaction (k2app) were found to be almost independent of the protein concentration with a mean value of 0.36 (s-1) (Fig. 5D)”. However, in Figure 5D, the value indicated corresponds to 0.38 (s-1). I kindly recommend that the Authors clarify this point.
(Answer and response) The value indicated in Fig. 5D is correct. In the revised manuscript, we changed as follows; “The apparent rate constants of the re-reduction reaction (k2app) were found to be almost independent of the protein concentration with a mean value of 0.38 (s-1) (Fig. 5D)”
5 - In line 645, where it reads “Fig. 12(A)” should be “Fig. S12(A)”.
(Answer and response) This was our mistake. “Fig. 12(A)” was changed to “Fig. S12(A)” in the revised manuscript.
6 - I kindly suggest that the Authors include a description of the data shown in Figure 8C at the end of the Results section, as it is not currently mentioned in the manuscript.
(Answer and response) We are very sorry for this mistake. We added the description of the data shown in Figure 8C in the Results section of the revised manuscript, as shown below (highlighted in yellow).
3.9. RNAi experiments: To confirm our hypothesis, we performed RNAi experiments. If Cecytb-2 protein indeed functions as a transmembrane ferric reductase in collaboration with SMF-3 protein for the acquisition of iron into the body of C. elegans, knock-down of Cecytb-2 gene expression (Fig. S13(A)) would cause a significant inhibition of the iron absorption ability and, accordingly, the growth of such iron-deficient worms would be depressed significantly [51]. However, for control N2 worms grown under normal NGM plate, knock-down of the Cecytb-2 gene by feeding RNAi method did not show any appreciable changes in their phenotypes, such as their life span (Fig. S13(B)), motility (Fig. S13(C)), memory (Fig. S13(D)), growth (Fig. 8A, left panels) and the number of hatched eggs (Fig. 8C).We thought that the knock-down worms could somehow maintain their normal growth by utilizing small but sufficient amounts of ferrous ions in their environments (i.e., in the NGM plates). To prove this possibility, we removed the environmental ferrous ions by using a specific ferrous ion chelator, BP (2,2’-dipyridyl) [51, 63] in the NGM plates with final concentrations of 20~60 μM. Addition of BP (60 μM) alone to the NGM plate did affect significantly on the number of hatched eggs from control N2 worms (Fig. 8C, left ), but it did not affect the growth of control N2 worms at all (Fig. S13 (B)). However, Cecytb-2-RNAi-treated N2 worms showed significant retardation in their growth (but not in their life span (Fig. S13 (B))) when they were laid on the NGM plate containing BP (40 μM) (Fig. 8A, left panels). Further, hatching of the eggs from Cecytb-2-RNAi-treated N2 worms did not occur by addition of BP (60 μM) (Fig. 8C, right). These results indicated that the Cecytb-2 knock-down worms could not utilize the environmental ferric ions sufficiently because of the suppression of the ferric reductase activities when they were laid on the NGM plates containing BP.
7 - The Authors suggest that the inhibition of electron transfer from AsA to the heme bL center upon DEPC treatment is primarily due to N-carbethoxylation of His101 and Lys98, as indicated by the MS/MS analysis. While this is a reasonable hypothesis, it remains suggestive. A practical approach to test it would be to generate a recombinant Cecytb-2 protein with these two residues mutated and evaluate it using the same assay. This would provide a direct and straightforward way to confirm the proposed mechanism.
(Answer and response) Thank you very much for valuable suggestions. Actually, we have done such studies using recombinant Cecytb-2 protein (for Lys98 and other residues, but not for His101, which caused the loss of heme b coordination). The results were consistent with our view and indicated that Lys98 has very important roles for the fast electron acceptance from AsA, as mentioned briefly in the Discussion section of the original version as (Fukuzawa et al., manuscript in preparation). We are currently preparing manuscripts to publish these new results.
8 - In line 878, please replace “purported” with “supported”.
(Answer and response) We changed to “supported” in the revised manuscript.